# TempSamp-R1: Effective Temporal Sampling with Reinforcement Fine-Tuning for Video LLMs

**Yunheng Li**[1]     **Jing Cheng**[2]     **Shaoyong Jia**[2]     **Hangyi Kuang**[1]
**Shaohui Jiao**[2]     **Qibin Hou**[1,3†]     **Ming-Ming Cheng**[1,3]
[1]VCIP, School of Computer Science, Nankai University,
[2]ByteDance Inc., [3]NKIARI, Futian, Shenzhen
yunhengli@mail.nankai.edu.cn

## Abstract

This paper introduces TempSamp-R1, a new reinforcement fine-tuning framework designed to improve the effectiveness of adapting multimodal large language models (MLLMs) to video temporal grounding tasks. We reveal that existing reinforcement learning methods, such as Group Relative Policy Optimization (GRPO), rely on on-policy sampling for policy updates. However, in tasks with large temporal search spaces, this strategy becomes both inefficient and limited in performance, as it often fails to identify temporally accurate solutions. To address this limitation, TempSamp-R1 leverages ground-truth annotations as off-policy supervision to provide temporally precise guidance, effectively compensating for the sparsity and misalignment in on-policy solutions. To further stabilize training and reduce variance in reward-based updates, TempSamp-R1 provides a non-linear soft advantage computation method that dynamically reshapes the reward feedback via an asymmetric transformation. By employing a hybrid Chain-of-Thought (CoT) training paradigm, TempSamp-R1 optimizes a single unified model to support both CoT and non-CoT inference modes, enabling efficient handling of queries with varying reasoning complexity. Experimental results demonstrate that TempSamp-R1 outperforms GRPO-based baselines, establishing new state-of-the-art performance on benchmark datasets: Charades-STA (R1@0.7: 52.9%, **+2.7**%), ActivityNet Captions (R1@0.5: 56.0%, **+5.3**%), and QVHighlights (mAP: 30.0%, **+3.0**%). Moreover, TempSamp-R1 shows robust few-shot generalization capabilities under limited data. Code is available at https://github.com/HVision-NKU/TempSamp-R1.

## 1   Introduction

Multimodal Large Language Models (MLLMs) [6, 21, 30, 31, 34, 40, 51, 58, 64] have demonstrated impressive capabilities in comprehending video content by following general human instructions and effectively interpreting visual content. However, their application to temporal video understanding tasks, such as temporal grounding [1, 22, 25, 35] and highlight detection [24, 43, 62], remains challenging, as these tasks require precise spatio-temporal understanding over long video sequences. A common approach is Supervised Fine-Tuning (SFT) [13, 17, 18, 28, 41, 45, 49, 54, 66], which aligns model predictions with static ground-truth timestamps using deterministic supervision. However, these methods often exhibit limited effectiveness, as models tend to overfit to deterministic timestamp supervision and fail to acquire the temporal reasoning required for flexible event localization [33, 60].

Recent approaches [10, 33, 60, 72] attempt to address these challenges using reinforcement learning (RL) frameworks. In particular, Group Relative Policy Optimization (GRPO) [47] improves temporal

---

†Corresponding author.

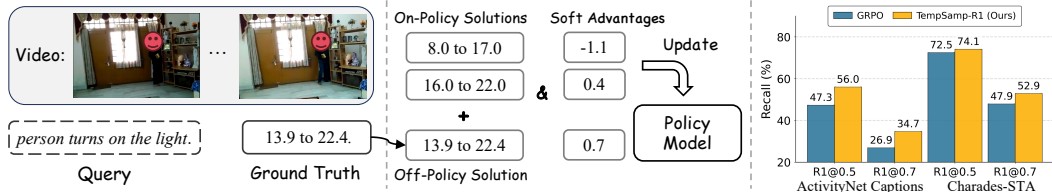

Figure 1: TempSamp-R1 integrates high-quality off-policy solutions with on-policy sampling, combined with soft advantage estimation to enable stable policy updates. It outperforms GRPO, which relies solely on on-policy sampling, on both Charades-STA and ActivityNet Captions.

performance by updating policies via grouped comparisons of sampled solutions, mitigating overfitting to static annotations. GRPO-based methods, such as TimeZero [60] and VideoChat-R1 [33], optimize models via task-specific rewards (e.g., temporal Intersection-over-Union (IoU)) to more effectively align visual dynamics with timestamped semantics in long videos. Despite these advancements, these methods still face a critical limitation: ***The vast temporal search space in temporal grounding tasks severely hinders effective exploration.*** This issue is empirically shown in Fig. 4, when employing GRPO, purely on-policy optimization leads to low and unstable top-1 IoU rewards, particularly on ActivityNet Captions, reflecting unstable early updates, ineffective learning under sparse supervision, and premature convergence to suboptimal solutions.

Notably, most video understanding datasets provide high-quality annotations (e.g., event timestamps) that offer precise supervision for grounding tasks. However, existing GRPO methods treat these annotations solely as evaluation (e.g., computing IoU) rather than dynamic learning sources, leading to suboptimal policy updates. Motivated by this observation, we propose TempSamp-R1, a new reinforcement finetuning framework that integrates on-policy generation with off-policy guidance to facilitate more stable and efficient policy optimization. As illustrated in Fig. 1, TempSamp-R1 incorporates high-quality and instruction-aligned solutions from external sources (e.g., ground truth annotations) as off-policy guidance, providing temporally precise supervision to compensate for the sparsity and misalignment often encountered in on-policy samplings.

However, since off-policy solutions are not sampled from the on-policy model, directly using their rewards can introduce substantial discrepancies in reward distribution, resulting in biased advantage estimation. This estimation bias may suppress high-quality on-policy samplings that diverge from the off-policy solutions, thereby limiting the policy's ability to generalize and explore alternative solutions effectively. To mitigate this, TempSamp-R1 introduces a non-linear soft advantage estimation mechanism inspired by the principles of adaptive reward shaping [15, 39]. To be specific, instead of treating all rewards uniformly, our method distinguishes the learning dynamics between high-reward and low-reward solutions by compressing the advantage values of near-optimal solutions and amplifying the relative reward gaps among suboptimal ones. This asymmetric shaping generates more informative gradients and facilitates stable policy refinement.

By incorporating a hybrid Chain-of-Thought (CoT) [61] training paradigm into a unified model, TempSamp-R1 achieves robust performance under both CoT and non-CoT reasoning modes, which are also shown to be complementary. We evaluate TempSamp-R1 through comprehensive experiments on temporal video understanding benchmarks, including temporal video grounding and video highlight detection. Extensive experiments show that TempSamp-R1 consistently outperforms various SFT-based and GRPO-based methods. Specifically, TempSamp-R1 improves temporal grounding recall accuracy on Charades-STA [11] (R1@0.7: 47.9% → 52.9%) and ActivityNet Captions [22] (R1@0.5: 50.7% → 56.0%), while enhancing highlight detection in QVHighlights [24] (mAP: 27.0% → 30.0%). Notably, TempSamp-R1 maintains competitive performance under limited supervision, highlighting its strong generalization capacity in few-shot scenarios.

## 2 Related Work

**Reinforcement finetuning**. Reinforcement learning has emerged as a powerful paradigm for enhancing the reasoning capabilities of large language models and MLLMs [7–9, 12, 19, 38, 52]. For instance, OpenAI's o1 model [19], DeepSeek-R1 [12] and Qwen3 [53] apply RL to generate intermediate reasoning traces before producing final responses, thereby improving performance on

complex reasoning tasks. Existing RL fine-tuning approaches can be broadly categorized into reward model-based methods and direct preference optimization techniques. Reward model-based methods, such as RLHF [4], PPO [46], and RLAIF [23], rely on a separately trained reward model to guide policy updates. In contrast, direct preference optimization methods, including DPO [44], IPO [2] and ORPO [16], bypass explicit reward modeling by optimizing preferences directly. GRPO [47] builds upon this paradigm by introducing on-policy sampling and groupwise preference evaluation, enabling dynamic policy refinement from richer comparative samplings. By leveraging comparisons between multiple candidate solutions, GRPO-based methods capture richer alignment supervision than standard SFT, which only learns from a single reference solution [27, 48, 50, 65, 70, 71]. Nonetheless, GRPO's performance still hinges on the diversity and informativeness of samplings, as uninformative comparisons may lead to optimization bias or degraded policy exploration.

**Temporal video grounding**. Temporal video understanding tasks, such as temporal grounding and highlight detection, require models to accurately identify and describe events within untrimmed video sequences [5, 20, 36, 37, 42, 67, 69, 73]. SFT has been the predominant approach for adapting MLLMs to these tasks [3, 32, 55–57, 74]. Models like TimeChat [45] employ video-text pre-training strategies to align frame-level features with textual descriptions. However, these methods struggle to achieve precise temporal localization due to the limitations in modeling long-range temporal dependencies and the tendency to rely on learned language patterns over visual cues. To address these challenges, RL has been introduced as a fine-tuning strategy to enhance the temporal reasoning capabilities of MLLMs. Recent methods, including TimeZero [60], R1-Omni [72], and VideoChat-R1 [33], utilize GRPO to fine-tune models on spatio-temporal perception tasks. These methods emphasize the design of reward functions to guide the model toward more accurate temporal localization. Our method also builds on the GRPO framework but differs from prior work in that it replaces random on-policy sampling with off-policy solutions to alleviate reward sparsity. To facilitate more efficient and stable policy optimization, we introduce a non-linear soft advantage estimation that dynamically reshapes advantage values to smooth gradient updates.

# 3 Methodology

To enable effective exploration beyond the limitations of on-policy learning, we propose TempSamp-R1, which combines on-policy generation with off-policy guidance and enhances training stability through soft advantage estimation. As shown in Fig. 2, our method introduces a mixed-policy training strategy based on GRPO, integrating high-quality external solutions (e.g., ground-truth annotations) into the policy optimization process. To stabilize training, we develop a soft advantage estimation mechanism that decouples and shapes reward to reduce gradient variance and adjust advantage bias, thereby promoting robust exploration and convergence.

## 3.1 Preliminaries

GRPO is a sample-efficient policy optimization algorithm designed to optimize policy models (e.g., MLLMs) by comparing groups of solutions, thereby eliminating the need for an independent value model and reducing computational overhead. Given a query $q$, GRPO samples a group of $G$ outputs $\{o_1, o_2, \ldots, o_G\}$ from the current policy model $\pi_\theta$ and computes the corresponding rewards $\{r_1, r_2, \ldots, r_G\}$ using a task-specific reward function that evaluates output quality with respect to ground-truth annotations and predefined specific rules (e.g., IoU). The advantage for each solution $o_i$ is then computed as $A_i = \frac{r_i - \mu}{\sigma}$, where $\mu$ and $\sigma$ denote the mean and standard deviation of the group rewards, respectively. This group-normalized advantage serves as the core optimization direction in GRPO, as it adaptively amplifies preferences for outputs that exhibit relatively higher quality within the sampled group. To update the policy, GRPO reuses the same solutions sampled from the previous policy $\pi_{\theta_{old}}$ and re-evaluates their likelihoods under the current policy $\pi_\theta$. The update applies importance weighting with a clipping mechanism and incorporates a KL-regularization term to constrain deviation from a reference policy $\pi_{ref}$:

$$\mathcal{J}_{GRPO}(\theta) = \frac{1}{G} \sum_{i=1}^{G} \left[ \min\left( \frac{\pi_\theta(o_i|q)}{\pi_{\theta_{old}}(o_i|q)} A_i, \text{clip}\left( \frac{\pi_\theta(o_i|q)}{\pi_{\theta_{old}}(o_i|q)}, 1-\epsilon, 1+\epsilon \right) A_i \right) - \beta \, \text{KL}(\pi_\theta || \pi_{ref}) \right], \quad (1)$$

Here, $\epsilon$ and $\beta$ denote the clipping range and the KL-divergence penalty weight, respectively. These components collectively impose constraints on policy updates and mitigate instability during the

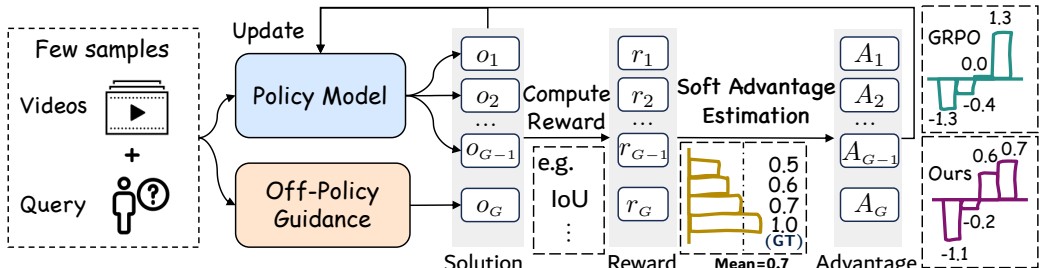

Figure 2: Overview of the TempSamp-R1 framework used to fine-tune the multimodal policy model. Given a few training examples, both the policy model and the off-policy guidance are used to generate solutions. Rewards are computed for each solution, and a soft advantage estimation module transforms raw rewards into standardized advantages for stable policy optimization. Right: Comparison of normalized advantages from GRPO (top) and our method (bottom), illustrating improved advantage discrimination. For clarity, the reference model and KL penalty are omitted.

optimization process. Recent methods [33, 60] often adopt $\pi_{\theta_{old}} = \pi_\theta$ to balance training data utilization efficiency and computational cost. Under this configuration, the importance weights collapse to unity, eliminating the need for ratio clipping while ensuring stable learning by updating the policy only once per sampling batch.

### 3.2 TempSamp-R1

Despite their impressive performance on general vision-language tasks, MLLMs often exhibit limited temporal grounding capabilities in video understanding [3, 56]. As a result, training under GRPO with on-policy sampling leads to slow convergence and a constrained performance upper bound, as the policy model encounters significant challenges in generating temporally precise solutions.

**Mix-policy sampling**. To address the above limitation, we introduce a mixed-policy training strategy that incorporates external off-policy solutions to provide accurate and query-specific temporal grounding. Although such high-quality solutions can be derived from expert policies, we adopt a more direct and empirically effective alternative by utilizing ground-truth annotations as the external policy guidance. To unify supervision across policy sources, we normalize the advantage values using the joint distribution of on-policy and off-policy rewards. Specifically, for each query, we sample $G-1$ solutions from the current policy and include one external off-policy solution (e.g., ground-truth). The normalized advantage for each solution with reward $r_i$ is then computed as:

$$A_i = \frac{r_i - \text{mean}(r_1, r_2, \ldots, r_{G-1} \cup r_G)}{\text{std}(r_1, r_2, \ldots, r_{G-1} \cup r_G)}, \tag{2}$$

where $r_G$ denotes the reward from the external off-policy solution. While the integration of off-policy solutions enhances the diversity and quality of training supervision, it also introduces skewed reward distributions that can destabilize policy optimization. In particular, when the off-policy solution consistently attains the highest reward, even marginal deviations among the remaining rewards (such as low inter-sample variance or tightly clustered suboptimal rewards) can render the normalization process numerically unstable. As illustrated in Fig. 2, the inclusion of off-policy solutions with exceptionally high rewards elevates the intra-group mean reward. In this scenario, the original GRPO algorithm computes negative advantages for all on-policy generated solutions, including those of high quality. This misestimation adversely affects advantage calculation, disrupts gradient updates, and diminishes exploration, ultimately causing premature convergence to suboptimal solutions. To mitigate these adverse effects and ensure stable and reliable advantage estimation, we propose three alternative strategies that regulate the contribution of off-policy supervision.

**Reward downscaling.** A straightforward mitigation strategy involves explicitly bounding the off-policy reward by scaling it to a fixed fraction (e.g., 80%) of the maximum possible value. This heuristic prevents off-policy solutions from dominating the advantage computation, mitigating distributional shift, and preserving gradient stability. However, the fixed nature of this scaling may suppress valuable learning supervision when off-policy solutions offer genuinely optimal samplings.

**Advantage anchoring**. To leverage off-policy supervision while mitigating distributional bias, we introduce an anchoring mechanism that decouples external solutions from on-policy advantage estimation. Specifically, off-policy samples are excluded from the computation of group statistics and do not participate in normalization. Instead, their advantage values are computed by scaling the maximum on-policy advantage within the group:

$$A_G = \lambda_{\text{off}} \cdot \max \left\{ A_i \mid i \in \{1, 2, \cdots, G-1\} \right\}, \tag{3}$$

where $\lambda_{\text{off}} = 1.2$ is a fixed scaling factor. This anchoring preserves the supervision from off-policy data while constraining its influence, maintaining stability and consistency in policy gradient updates.

**Non-linear reward shaping**. To improve stability under skewed reward distributions, we apply a non-linear transformation to the rewards prior to advantage computation. This transformation is defined as an asymmetric piecewise function that compresses high rewards and expands low rewards. The shaping reward $\tilde{r}_i$ is defined as:

$$\tilde{r}_i = \begin{cases} \tau + \alpha_1 \cdot \ln\left((r_i - \tau) + 1\right), & r_i \geq \tau \\ \tau - \dfrac{e^{\alpha_2 \cdot (\tau - r_i)} - 1}{e^{\alpha_2} - 1}. & r_i < \tau \end{cases} \tag{4}$$

Here, $\tau = 0.8$ is the reward threshold, $\alpha_1 = 0.01$ controls compression above the threshold, and $\alpha_2 = 1$ governs expansion below it. The logarithmic branch mitigates gradient spikes from optimal solutions, while the exponential branch increases contrast among suboptimal samplings.

Each of the above strategies provides an alternative mechanism to mitigate instability caused by incorporating strong off-policy solutions. By independently adjusting rewards, advantage scales, or reward distributions, these methods offer flexible design choices to control the influence of external off-policy and improve the stability of policy optimization in multimodal video understanding.

## 3.3 Training

We adopt a two-phase training scheme with task-specific reward functions to support diverse video understanding tasks. In the initialization phase, the model is optimized to generate accurate final answers without explicit reasoning. Building on this, we incorporate format rewards to encourage the generation of intermediate reasoning steps alongside final outputs. To operationalize this strategy across different tasks, we define a suite of task-specific reward functions that directly guide policy optimization toward task-relevant behaviors.

**IoU reward**. For temporal localization tasks, the model is required to predict an event interval $[t_{\text{p}}^{\text{s}}, t_{\text{p}}^{\text{e}}]$ conditioned on a given query. To quantify prediction accuracy, we define a reward rule based on the IoU with the ground truth interval $[t_{\text{g}}^{\text{s}}, t_{\text{g}}^{\text{e}}]$, computed as $R_{\text{IoU}} = (\min(t_p^e, t_g^e) - \max(t_p^s, t_g^s))/(\max(t_p^e, t_g^e) - \min(t_p^s, t_g^s))$. The $R_{\text{IoU}}$ is computed as the ratio between the intersection and the union of the two intervals, serving as a direct measure of temporal alignment accuracy.

**Timestamp matching reward**. For highlight detection tasks, the model jointly predicts temporal boundaries and associated saliency scores. To evaluate both the structural and semantic quality of these predictions, we define a composite reward: $R_{\text{ts}} = \lambda_{\text{rec}} \cdot \text{F2} + \lambda_{\text{score}} \cdot \frac{1}{1+\text{WMSE}}$. Here, the F2 score measures the temporal alignment by computing a recall-weighted F-measure over matched timestamps, emphasizing recall to better capture relevant highlights. The Weighted Mean Squared Error (WMSE) assesses the fidelity of predicted saliency scores, with weights derived from the squared ground-truth scores to emphasize high-saliency regions. We set $\lambda_{\text{rec}} = 0.6$ and $\lambda_{\text{score}} = 0.4$ to prioritize semantic fidelity in salient regions while ensuring temporal coverage.

**Format reward**. To promote structured output in reasoning tasks, we introduce a format reward that enforces conformity to a predefined schema. The model is expected to generate reasoning enclosed in `<Think>...</Think>` and final answers in `<Answer>...</Answer>`. The reward is set to 1 if the output matches the required structure based on regular expression validation, and 0 otherwise.

# 4 Experiments

**Implementation details.** Our experiments are conducted using the Qwen2.5-VL-7B-Instruct model [3]. To ensure a fair comparison with prior efficient video fine-tuning methods [33, 60],

| Method | Type | Charades-STA | | | | ActivityNet Captions | | | | QVHighlights | |
|---|---|---|---|---|---|---|---|---|---|---|---|
| | | mIoU | R1@0.3 | R1@0.5 | R1@0.7 | mIoU | R1@0.3 | R1@0.5 | R1@0.7 | mAP | HIT@1 |
| *Supervised Fine-Tuning (SFT) Methods* | | | | | | | | | | | |
| UnLoc-L [63] | SFT | - | - | 60.8 | 38.4 | - | - | 48.3 | 30.2 | - | - |
| Timechat [45] | SFT | - | - | 46.7 | 23.7 | - | - | - | - | 21.7 | 37.9 |
| HawkEye [59] | SFT | 49.3 | 72.5 | 58.3 | 28.8 | 39.1 | 55.9 | 34.7 | 17.9 | - | - |
| TRACE [14] | SFT | - | - | 61.7 | 41.4 | - | - | 37.7 | 24.0 | - | - |
| VideoChat-T [68] | SFT | - | 79.4 | 67.1 | 43.0 | - | - | - | - | 27.0 | 55.3 |
| iMOVE [29] | SFT | 57.9 | 79.8 | 68.5 | 45.3 | 49.3 | 67.2 | 50.7 | 32.4 | - | - |
| *Reinforcement Learning (RL) Methods based on Qwen2.5-VL-7B* | | | | | | | | | | | |
| Qwen2.5-VL-7B [3]* | - | 49.7 | 73.4 | 54.4 | 30.3 | 33.1 | 45.2 | 29.7 | 18.1 | 19.7 | 34.1 |
| VideoChat-R1 [33] | RL | 60.8 | - | 71.7 | 50.2 | - | - | - | - | - | - |
| VideoChat-R1-thinking [33] | RL | 59.9 | - | 70.6 | 47.2 | - | 68.6 | 47.3 | 26.9 | - | - |
| TimeZero [60] | RL | - | 83.3 | 72.5 | 47.9 | - | 68.6 | 47.3 | 26.9 | - | - |
| **TempSamp-R1**(no-CoT) | RL | 61.7 | **83.3** | 73.6 | 52.2 | 52.1 | 72.8 | 55.4 | 34.2 | **30.0** | **57.6** |
| **TempSamp-R1** (CoT) | RL | **62.1** | 83.6 | **74.1** | **52.9** | **52.4** | **73.4** | **56.0** | **34.7** | 28.3 | 54.9 |
| TempSamp-R1 Mixed CoT | RL | 64.2 | 85.0 | 76.0 | 56.3 | 54.9 | 75.7 | 58.7 | 37.6 | 29.3 | 63.7 |

Table 1: Performance comparison of Charades-STA, ActivityNet Captions, and QVHighlights datasets. Our TempSamp-R1 supports both CoT and no-CoT reasoning within a single unified model, and achieves strong performance across all datasets. The TempSamp-R1 Mixed CoT selects the better prediction between CoT and no-CoT for each query.

| Method | mIoU | R1@0.3 | R1@0.5 | R1@0.7 |
|---|---|---|---|---|
| SFT | 20.6 | 30.2 | 16.7 | 7.9 |
| GRPO | 30.7 | 45.0 | 27.5 | 12.9 |
| TempSamp-R1 | **34.7** | **50.9** | **32.2** | **16.2** |

Table 2: Out-of-domain generalization performance from Charades-STA to ActivityNet.

we standardize the input preprocessing pipeline. Specifically, all videos are temporally downsampled to 2 frames per second (FPS) and resized to approximately 2.8 million pixels per frame. Training is performed on four NVIDIA A100 GPUs with a batch size of 1 per GPU. For GRPO-based training, each question is associated with a total of 4 solutions, consisting of $G=3$ on-policy samplings and 1 off-policy solution. This setting balances computational efficiency and diversity in policy learning. For Charades-STA (Tab. 1), we increase the number of solutions to 8 to better accommodate the task's higher compositional complexity.

**Benchmarks.** We evaluate our model on the temporal grounding task using the Charades-STA, ActivityNet Captions, and QVHighlights datasets. Following established practices [45, 68], we report Recall@1 (R1@) at Intersection over Union (IoU) thresholds of 0.3, 0.5, and 0.7. Additionally, we compute the mean IoU (mIoU) across all test samples to evaluate overall localization accuracy. The QVHighlights dataset evaluates using mean Average Precision (mAP) at IoU thresholds of 0.5 and 0.75, and HIT@1, which indicates whether the top-ranked clip is labeled as "Very Good."

## 4.1 Main results

**Fine-tuning performance.** We evaluate the effectiveness of our proposed method, TempSamp-R1, across three standard benchmarks for temporal grounding: Charades-STA, ActivityNet Captions, and QVHighlights. Tab. 1 compares our approach with a wide range of baselines, including zero-shot, supervised fine-tuning (SFT) methods, and reinforcement learning (RL) based approaches. Compared to state-of-the-art SFT baselines, TempSamp-R1 achieves stronger performance. On Charades-STA, it obtains 74.1% R1@0.5 and 52.9% R1@0.7, outperforming iMOVE by +5.6% and +7.6% respectively. In comparison to existing RL-based approaches, TempSamp-R1 achieves consistent gains across all benchmarks. For instance, it surpasses VideoChat-R1 by +2.4% R1@0.5 and TimeZero by +5.0% R1@0.7 on Charades-STA. On ActivityNet Captions, it exceeds TimeZero by +8.7% R1@0.5 and

---

*For Qwen2.5-VL-7B, the results on Charades-STA and ActivityNet Captions are reproduced from [26], whereas the results on QVHighlights are obtained from our implementation.

| Method | 50 videos | | 100 videos | | 200 videos | | 500 videos | | |
| --- | --- | --- | --- | --- | --- | --- | --- | --- | --- |
| | R1@0.5 | mIoU | R1@0.5 | mIoU | R1@0.5 | mIoU | R1@0.5 | mIoU | Training Time |
| SFT | 44.8 | 41.9 | 46.5 | 42.6 | 45.2 | 42.7 | 51.4 | 46.2 | 93 min |
| GRPO | 36.2 | 38.4 | 39.3 | 40.8 | 43.5 | 43.8 | 55.3 | 49.8 | 338 min |
| TempSamp-R1 (Ours) | **46.7** | **44.7** | **54.0** | **49.1** | **58.2** | **51.8** | **64.0** | **55.1** | 218 min |

Table 3: Few-shot performance comparison of SFT, GRPO, and our proposed method on Charades-STA under varying training sample sizes (50, 100, 200, 500). All models were trained for 3 epochs.

| Method | R1@0.3 | R1@0.5 | R1@0.7 |
| --- | --- | --- | --- |
| GRPO (baseline) | 81.2 | 68.9 | 46.0 |
| Mixed-policy | 77.8 | 63.0 | 41.3 |
| Reward downscaling | 81.2 | 70.3 | 48.1 |
| Advantage anchoring | 81.8 | 70.7 | 49.1 |
| Non-linear reward shaping | **82.9** | **72.1** | **49.6** |

Table 4: Ablation results comparing GRPO with enhanced variants incorporating mixed-policy rewards and alternative advantage shaping strategies.

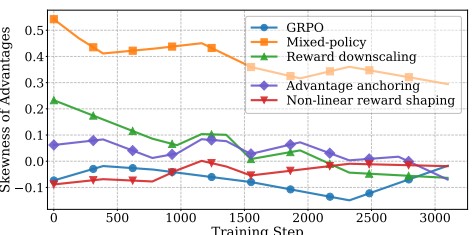

Figure 3: Skewness of the advantage distributions during training for different variants.

+7.8% R1@0.7. We observe that CoT prompting at inference consistently improves performance on Charades-STA and ActivityNet Captions, indicating that explicit reasoning is beneficial for tasks involving complex temporal dependencies. In contrast, on QVHighlights, the TempSamp-R1 with no-CoT performs better, indicating that direct prediction is more suitable for highlight detection, where explicit reasoning may be redundant or distracting. We further explore a mixed variant, TempSamp-R1 Mixed CoT, which selects the better output between CoT and no-CoT predictions for each query. This strategy consistently outperforms either individual reasoning mode, underscoring their complementary roles. Representative examples in Fig. 6 illustrate how each reasoning mode excels under different semantic and temporal conditions.

**Out-of-domain generalization.** To assess the cross-dataset transferability of different approaches, we conduct out-of-domain evaluations where all models are trained on Charades-STA and directly tested on ActivityNet Captions. As shown in Tab. 2, TempSamp-R1 consistently outperforms both SFT and GRPO across all metrics on both datasets. On ActivityNet Captions, it achieves improvements of +4.0% mIoU and +4.7% R1@0.5 over GRPO. These results suggest that off-policy supervision and soft advantage shaping jointly enhance the cross-domain transferability of TempSamp-R1.

**Few-shot performance.** We evaluate our method under few-shot settings on the Charades-STA dataset, training with 50, 100, 200, and 500 videos, each for 3 epochs. Table 3 presents the performance comparison among SFT, GRPO, and our proposed method. Our method consistently outperforms both SFT and GRPO across all training sizes. Notably, with just 50 training samples, our method achieves a mIoU of 44.7%, surpassing SFT by +2.8%. As the number of training samples increases, the performance gap widens. With 500 samples, our method attains an R1@0.5 of 64.0%, outperforming SFT by +12.6%, and GRPO by +8.7%, respectively. In terms of training efficiency, our method requires 218 minutes for training with 500 samples, which less than GRPO's 338 minutes. These results demonstrate that our method maintains efficient training, highlighting its practicality for real-world applications, where annotated data is limited.

## 4.2 Component-wise analysis of TempSamp-R1

We analyze our method on Charades-STA and ActivityNet Captions to assess the impact of key components in TempSamp-R1, including off-policy guidance and advantage shaping. Results show that these mechanisms jointly contribute to more stable training and better policy optimization.

**Analysis of advantage shaping strategies.** We analyze the advantage shaping strategies introduced in Sec. 3.2 through systematic ablations within the TempSamp-R1 framework here. Tab. 4 compares the GRPO baseline against four variants: mixed-policy supervision, reward downscaling, advantage anchoring, and non-linear reward shaping. Directly injecting ground-truth rewards (mixed-policy)

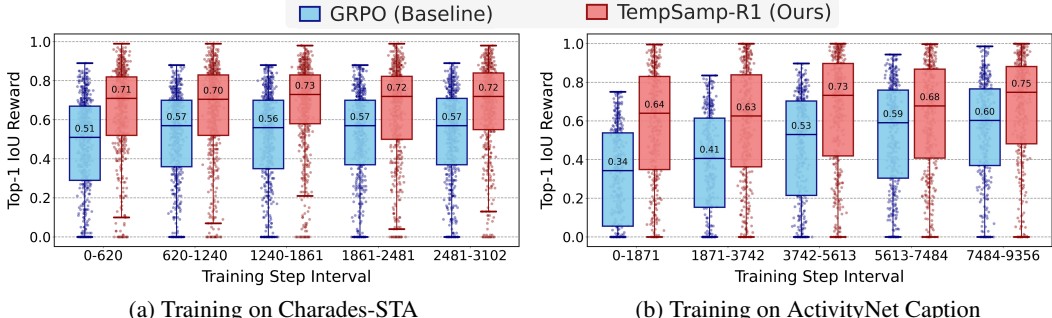

(a) Training on Charades-STA  (b) Training on ActivityNet Caption

Figure 4: Distribution of top-1 IoU rewards under GRPO and TempSamp-R1 on Charades-STA and ActivityNet Captions. TempSamp-R1 exhibits higher median rewards and reduced variance, indicating more stable and effective policy learning.

| Samplings | IoU=0.3 | | IoU=0.5 | | IoU=0.7 | |
|---|---|---|---|---|---|---|
| | GRPO | TempSamp-R1 ($+\Delta$) | GRPO | TempSamp-R1 ($+\Delta$) | GRPO | TempSamp-R1 ($+\Delta$) |
| 2 | 77.8 | 80.8 (+3.0) | 60.2 | 67.3 (+7.1) | 34.4 | 44.6 (+10.2) |
| 4 | 81.0 | 82.5 (+1.5) | 68.9 | 71.6 (+2.7) | 46.0 | 49.6 (+3.6) |
| 6 | 81.3 | 82.6 (+1.3) | 69.2 | 71.7 (+2.5) | 49.2 | 50.3 (+1.1) |
| 8 | 81.2 | 82.6 (+1.4) | 70.5 | 71.9 (+1.4) | 47.1 | 50.8 (+3.7) |

Table 5: Ablation study on the number of solutions. Our method consistently outperforms the baseline GRPO across all configurations, particularly under higher IoU thresholds.

leads to degraded performance (e.g., 63.0 R1@0.5), which can be attributed to distributional shift and reduced on-policy sampling diversity. Reward downscaling and advantage anchoring partially mitigate this issue, improving R1@0.5 to 70.3 and 70.7, respectively. Non-linear reward shaping achieves the best results. To better understand these results, Fig. 3 analyzes the skewness of the advantage distributions throughout training. GRPO exhibits persistent negative skewness, indicating dominance of low-reward solutions and weak gradient magnitude. In contrast, mixed-policy supervision results in high positive skewness, reflecting over-reliance on a small set of high-reward solutions and poor policy generalization. Our proposed three shaping strategies mitigate these imbalances to varying degrees. Notably, non-linear reward shaping maintains near-zero skewness throughout training, promoting stable optimization and improved grounding performance.

**Analysis of reward distribution.** To better understand the learning dynamics of different methods, we analyze the distribution of top-1 IoU rewards collected during training. Fig. 4 presents box plots comparing GRPO and methodname on Charades-STA and ActivityNet Captions. GRPO baseline exhibits low median rewards with high variance, especially on ActivityNet Captions, reflecting unstable exploration and frequent convergence to suboptimal solutions. In contrast, methodname yields significantly higher median rewards and notably reduced dispersion. This indicates that the integration of off-policy supervision and non-linear advantage shaping not only improves reward magnitude but also enhances training stability. The compact distribution further suggests that methodname can consistently identify high-quality solutions across different video samples, even in the presence of long temporal sequences and ambiguous queries.

### 4.3 Ablation study

We conduct ablations to evaluate the individual contributions of sampling strategy and CoT supervision. Results on Charades-STA show that TempSamp-R1 remains effective under limited on-policy samplings, and CoT supervision enhances temporal localization.

**Impact of the number of solutions.** We conduct an ablation study to evaluate how varying the number of solutions per query affects temporal grounding performance. Specifically, we assessed models with 2, 4, 6, and 8 solutions on the Charades-STA dataset. As shown in Tab. 5, TempSamp-R1 outperforms the baseline GRPO across all configurations. Notably, with only 2 solutions, TempSamp-R1 achieves a remarkable 10.2% absolute improvement in R1@0.7 over GRPO, demonstrating its robustness

| Method | Epochs | Training Prompt Think | Training Prompt Answer | Test Prompt Think | Test Prompt Answer | mIoU | R1@0.3 | R1@0.5 | R1@0.7 |
|---|---|---|---|---|---|---|---|---|---|
| Qwen2.5-VL-7B | - | - | - |  | ✓ | 29.0 | 44.7 | 24.2 | 11.1 |
| (baseline) | - | - | - | ✓ | ✓ | 28.1 | 41.8 | 23.4 | 11.1 |
| + TempSamp-R1 | 1 |  | ✓ |  | ✓ | 61.1 | 82.6 | 71.9 | 50.8 |
|  | 1 |  | ✓ | ✓ | ✓ | 54.2 | 75.9 | 63.6 | 40.5 |
|  | 2 |  | ✓ |  | ✓ | 61.5 | 82.6 | 73.2 | 51.2 |
|  | 2 |  | ✓ | ✓ | ✓ | 61.2 | 82.4 | 72.8 | 50.6 |
|  | 2 | ✓ | ✓ |  | ✓ | 61.8 | 83.3 | 73.6 | 52.2 |
|  | 2 | ✓ | ✓ | ✓ | ✓ | 62.1 | 83.6 | 74.1 | 52.9 |

Table 6: Ablation study on the effects of `<Think>` and `<Answer>` prompts during training and testing under various configurations of our method on the Charades-STA dataset.

| $\tau$ | R1@0.3 | R1@0.5 | R1@0.7 |
|---|---|---|---|
| 0.7 | 81.6 | 70.5 | 48.4 |
| 0.8 | **82.9** | **72.1** | **49.6** |
| 0.9 | 81.1 | 69.4 | 47.5 |

(a) Ablation on $\tau$.

| $\alpha_1$ | R1@0.3 | R1@0.5 | R1@0.7 |
|---|---|---|---|
| 0.05 | 81.6 | 70.5 | 48.4 |
| 0.01 | **82.9** | **72.1** | **49.6** |
| 0.02 | 81.9 | 71.0 | 49.1 |

(b) Ablation on $\alpha_1$.

| $\alpha_2$ | R1@0.3 | R1@0.5 | R1@0.7 |
|---|---|---|---|
| 0.5 | 82.2 | 71.6 | 49.0 |
| 1.0 | **82.9** | **72.1** | **49.6** |
| 2.0 | 81.1 | 69.5 | 46.9 |

(c) Ablation on $\alpha_2$.

Figure 5: Ablation study on the effects of different hyperparameters $\tau$, $\alpha_1$, and $\alpha_2$ in the non-linear reward shaping method.

under limited on-policy sampling. While increasing the number of solutions generally leads to improved performance for both methods, the gains for GRPO diminish beyond 4 solutions, whereas TempSamp-R1 continues to benefit from additional on-policy samplings.

**Impact of CoT supervision and inference.** We perform an ablation study to assess the impact of incorporating `<Think>` and `<Answer>` prompts during training and testing on the Charades-STA dataset. Tab. 6 presents the results across different combinations of training-phase supervision and inference-time prompting. The best performance is achieved when utilizing both `<Think>` and `<Answer>` prompts during training and inference. This suggests that the combination of CoT prompts enhances the model's temporal grounding capabilities. In contrast, applying `<Think>` prompts only during inference, without corresponding supervision during training, results in a substantial performance drop, suggesting sensitivity to mismatched prompting conditions. In addition, training solely with `<Answer>` prompts for two epochs results in reasonable performance but does not match the effectiveness of the hybrid CoT strategy, underscoring the benefits of incorporating CoT prompts during training. Notably, Tab. 6 (last rows) indicate that training with one non-CoT epoch followed by one CoT epoch yields stable performance regardless of test-time prompt design, indicating the model's robustness to CoT prompt variations after hybrid CoT training.

**Impact of non-linear reward shaping parameters.** We analyze how each parameter in the non-linear reward shaping method influences learning dynamics, as shown in Tab. 5. The threshold $\tau$ defines the boundary between high- and low-reward regions. Moderate values maintain balanced optimization, whereas larger $\tau$ exacerbate reward imbalance by concentrating rewards on a few high-value solutions, which reduces the model's ability to learn from informative suboptimal solutions. The coefficient $\alpha_1$ controls logarithmic scaling above the threshold. Smaller values retain gradient diversity, while larger values overly smooth the reward distribution. In contrast, $\alpha_2$ governs exponential scaling below the threshold. It enhances contrast among lower rewards but can cause instability when set too high. Overall, these findings indicate that the proposed non-linear shaping method is stable within a broad parameter range, with $\tau = 0.8$, $\alpha_1 = 0.01$, and $\alpha_2 = 1.0$ yielding the trade-off between reward contrast and optimization stability.

## 4.4 Qualitative results

To further illustrate the effectiveness of our method, we present qualitative examples from the Charades-STA and ActivityNet Captions datasets, as shown in Fig. 6. Compared to the GRPO baseline, our model achieves more accurate temporal localization and generates more coherent event reasoning, particularly in challenging or ambiguous cases. A key strength of TempSamp-R1 is its

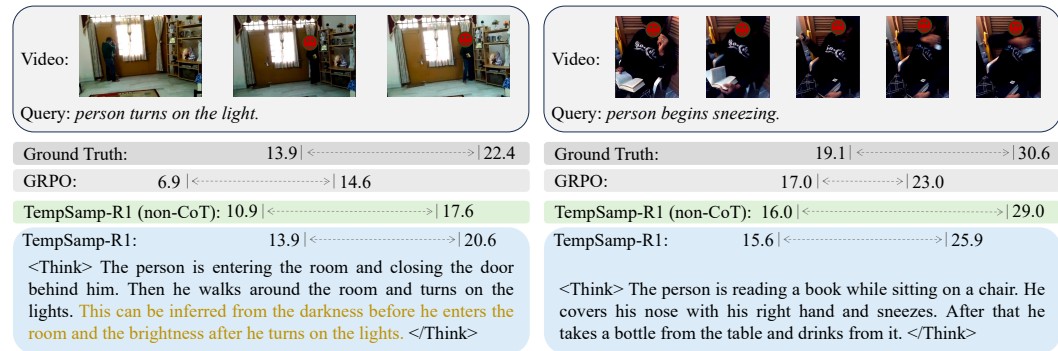

(a) Qualitative examples where TempSamp-R1 successfully queries. CoT inference shows stronger contextual reasoning for complex queries, while non-CoT mode yields sharper boundaries for straightforward actions.

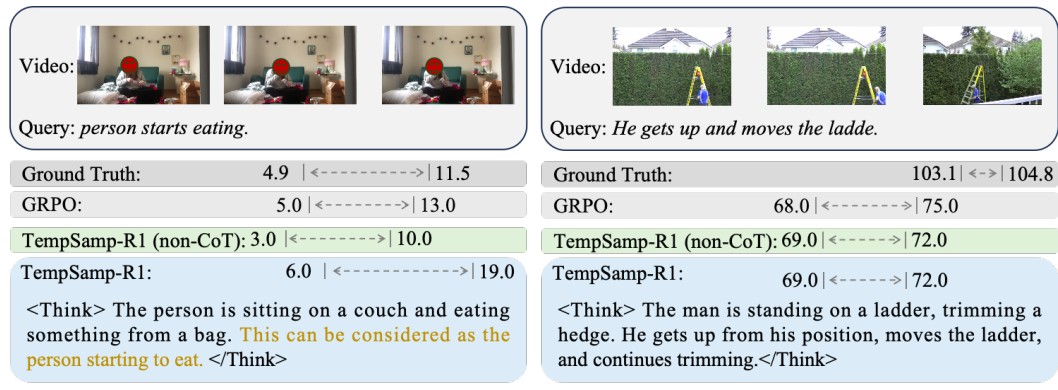

(b) Failure cases mainly arise from ambiguous annotations and subtle cues that challenge accurate localization.

Figure 6: Qualitative comparisons on temporal grounding using TempSamp-R1. TempSamp denotes evaluation without CoT, while TempSamp-R1 uses the `<Think>` prompt.

ability to optionally include the `<Think>` prompt at inference time. In cases requiring reasoning, such as detecting transitions based on subtle visual cues, the CoT prompt improves boundary precision. For instance, when identifying the moment a person turns on a light, the model correctly associates the event with a change in brightness. In contrast, for queries with clear visual markers, the model performs well without reasoning prompts, demonstrating adaptability to different reasoning demands.

However, as shown in the failure cases, errors often stem from ambiguous or narrowly defined ground-truth annotations, which introduce uncertainty in the evaluation. Additionally, some complex queries involve subtle or overlapping visual cues that challenge precise localization, suggesting potential areas for further improvement in both annotation quality and model reasoning.

## 5    Conclusions

This paper introduces TempSamp-R1, a reinforcement learning framework designed to enhance temporal video understanding in multimodal large language models. TempSamp-R1 integrates off-policy supervision with a non-linear soft advantage mechanism to address the challenges of sparse rewards and unstable policy updates inherent in large temporal search spaces. By promoting effective policy updates, TempSamp-R1 enables more accurate temporal localization of target events described in the input queries. Extensive evaluations on benchmarks such as Charades-STA, ActivityNet Captions, and QVHighlights demonstrate that TempSamp-R1 consistently outperforms both SFT-based and existing GRPO-based reinforcement learning methods.

**Acknowledgment.** This work was supported by NSFC (No. 62495061 and No. 62225604), the Shenzhen Science and Technology Program under Grant No. JCYJ20240813114237048 and the Fundamental Research Funds for the Central Universities (Nankai University) under Grant 070-63253220.

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

# Appendix

## A  Limitations and broader impact

**Limitations.** While TempSamp-R1 demonstrates consistent improvements over existing GRPO-based methods across multiple temporal grounding benchmarks, it also presents several limitations. First, the framework currently relies on the availability of high-quality off-policy supervision (e.g., ground-truth timestamps), which may not be accessible in weakly labeled scenarios. Second, although TempSamp-R1 is evaluated on temporal grounding and highlight detection tasks, its effectiveness on other video reasoning tasks (e.g., multi-event tracking) remains to be explored.

**Broader Impact.** This work contributes to the advancement of reinforcement learning in long video understanding. By combining on-policy exploration with structured off-policy guidance, TempSamp-R1introduces a more stable and data-efficient fine-tuning paradigm for vision-language models. We believe this direction can benefit downstream applications such as video retrieval, surveillance, and assistive robotics, where precise temporal reasoning is critical. Since our method is not designed for a specific application domain, it does not directly raise immediate societal or ethical concerns.

## B  Training details

We fine-tune our method on the Qwen2.5-VL-7B-Instruct model [3] using full-parameter optimization. Tab. 7 summarizes the full set of training configurations. The first-stage fine-tuning uses the same in-domain datasets as the second stage, matching the corresponding evaluation benchmarks (e.g., Charades-STA and QVHighlights).

| Trainable | Full Model |
|---|---|
| Per-device Batch Size | 1 |
| Gradient Accumulation | 2 |
| Epoch | 2 |
| Optimizer | AdamW |
| Deepspeed | Zero3-Offload |
| LR Schedule | $1 \times 10^{-6}$ |
| Max Generated Sequence Length | 2048 |
| GPU Nums | 4 A100 (80 GB) |

Table 7: Training configuration for TempSamp-R1.

## C  Additional Experiments

**Comparison of GRPO and TempSamp-R1 under CoT and non-CoT prompting** We compare the performance of GRPO and TempSamp-R1 with both CoT and non-CoT prompting strategies on the Charades-STA dataset. As shown in Tab. 8, TempSamp-R1 consistently outperforms GRPO regardless of the prompting approach. Notably, the improvements achieved with CoT prompting are more pronounced, with gains of 1.5 and 4.8 points at R1@0.5 and R1@0.7 respectively, compared to 1.4 and 3.7 points under non-CoT prompting. These results suggest that the stabilized training and reward optimization in TempSamp-R1 better complement CoT prompting, enabling more effective utilization of intermediate reasoning steps. This highlights the complementary benefits of advanced reward design and CoT prompting in improving temporal grounding.

## D  Visualizations

We present additional qualitative results to illustrate the behavior of TempSamp-R1 on temporal grounding tasks. Fig. 7 shows representative success cases from both Charades-STA and ActivityNet Captions, where TempSamp-R1 accurately localizes the queried events and generates coherent reasoning under both CoT and non-CoT settings.

In contrast, Fig. 8 highlights common failure cases, which primarily stem from ambiguous visual content, repeated actions, or loosely defined ground-truth annotations. tends to localize the earliest

| Method | Training Prompt | Test Prompt | R1@0.3 | R1@0.5 | R1@0.7 |
|--------|-----------------|-------------|--------|--------|--------|
| GRPO | Non-CoT | Non-CoT | 81.2 | 70.5 | 47.1 |
| TempSamp-R1 | Non-CoT | Non-CoT | **82.6** | **71.9** | **50.8** |
| GRPO | CoT | CoT | 83.0 | 72.6 | 48.1 |
| TempSamp-R1 | CoT | CoT | **83.6** | **74.1** | **52.9** |

Table 8: Performance comparison of GRPO and TempSamp-R1 with and without CoT prompting on Charades-STA dataset.

matching interval, whereas the annotation provides only a single reference segment, leading to apparent discrepancies. These examples further underscore the challenges of precise temporal localization.

# E  Datasets of training and evaluation

We train and evaluate TempSamp-R1 on three widely-used video-language datasets, spanning two representative tasks: temporal grounding and highlight detection.

**Charades-STA** [11] is a benchmark for temporal localization in indoor videos. Each sample consists of a natural language query and its corresponding temporal segment within a video. We follow standard splits with approximately 12.4k training and 3.7k validation examples.

License: Non-commercial use license provided by the Allen Institute for AI. https://prior.allenai.org/projects/charades

URL: https://github.com/jiyanggao/TALL

**ActivityNet Captions** [22] provides temporally grounded captions for diverse videos. Each video contains multiple event annotations with rich semantic content. The dataset comprises approximately 20K long, untrimmed videos, with an average of 3.65 sentence-event pairs per video. Following standard practice, we adopt the official dataset splits, resulting in 37,421 training, 17,505 validation, and 17,031 test samples. Compared to Charades-STA, this dataset covers a broader domain and more diverse activity types.

URL: https://cs.stanford.edu/people/ranjaykrishna/densevid/

**QVHighlights** [24] is a large-scale benchmark for evaluating query-conditioned highlight detection in long-form videos. The dataset comprises 10,148 curated video clips, each with a fixed duration of 150 seconds. Each video is paired with at least one user-issued natural language query describing salient moments, resulting in 10,310 unique queries and 18,367 annotated highlights. On average, each highlight spans approximately 24.6 seconds.

License: Attribution-NonCommercial-ShareAlike 4.0 International.

URL: https://github.com/jayleicn/moment_detr/tree/main/data

# F  Prompts

We provide prompt templates used for temporal video grounding and video highlight detection tasks under both CoT and non-CoT settings. For CoT prompts, models are instructed to first reason step-by-step within <think> tags before outputting final predictions in <answer> tags. Non-CoT prompts, by contrast, elicit direct answers without intermediate reasoning. All prompts are designed to standardize response formats and support consistent evaluation across different tasks and inference modes. See Fig. 9 for examples.

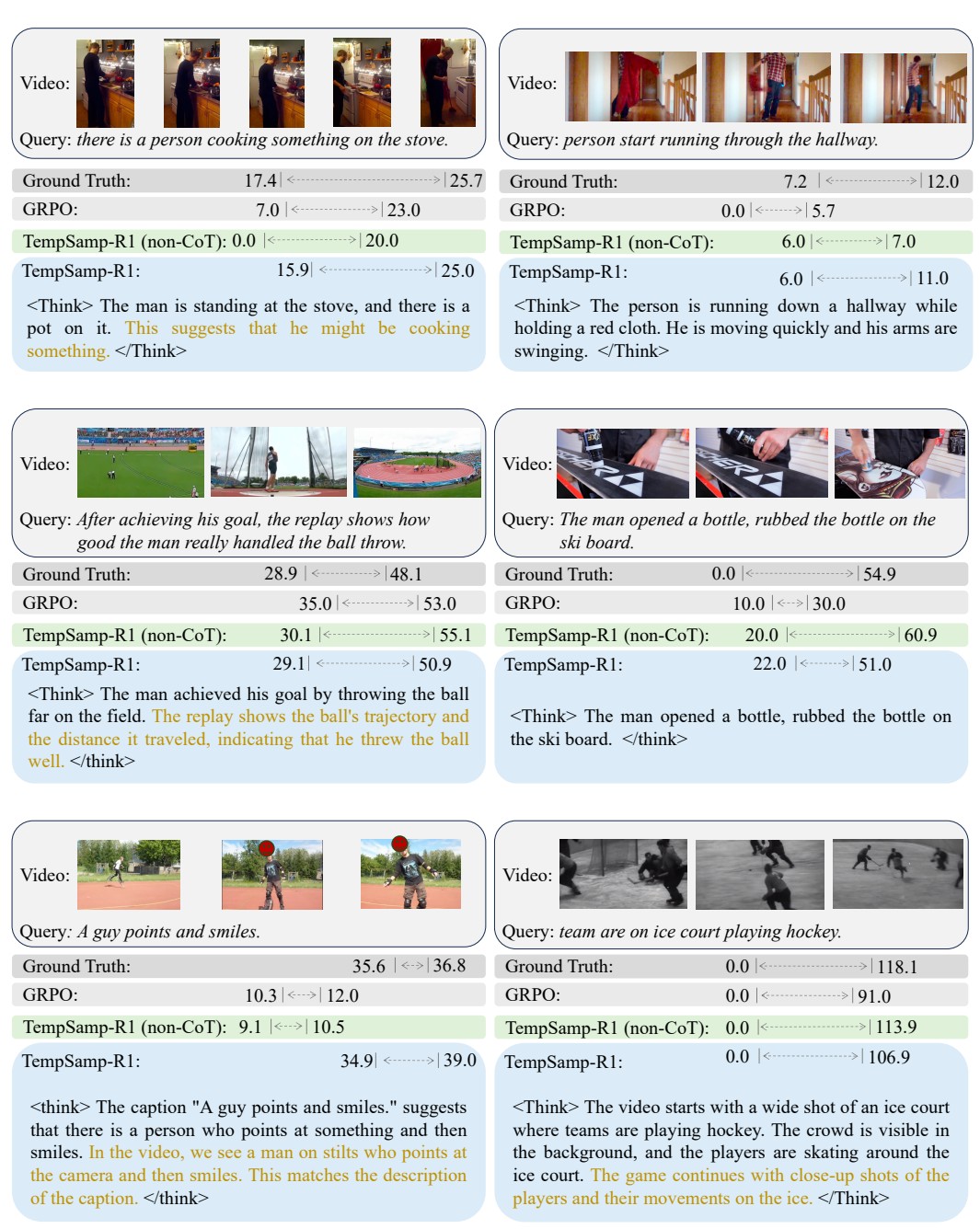

Figure 7: Qualitative comparisons on temporal grounding on Charades-STA and ActivityNet Captions. TempSamp-R1 accurately localizes the queried events.

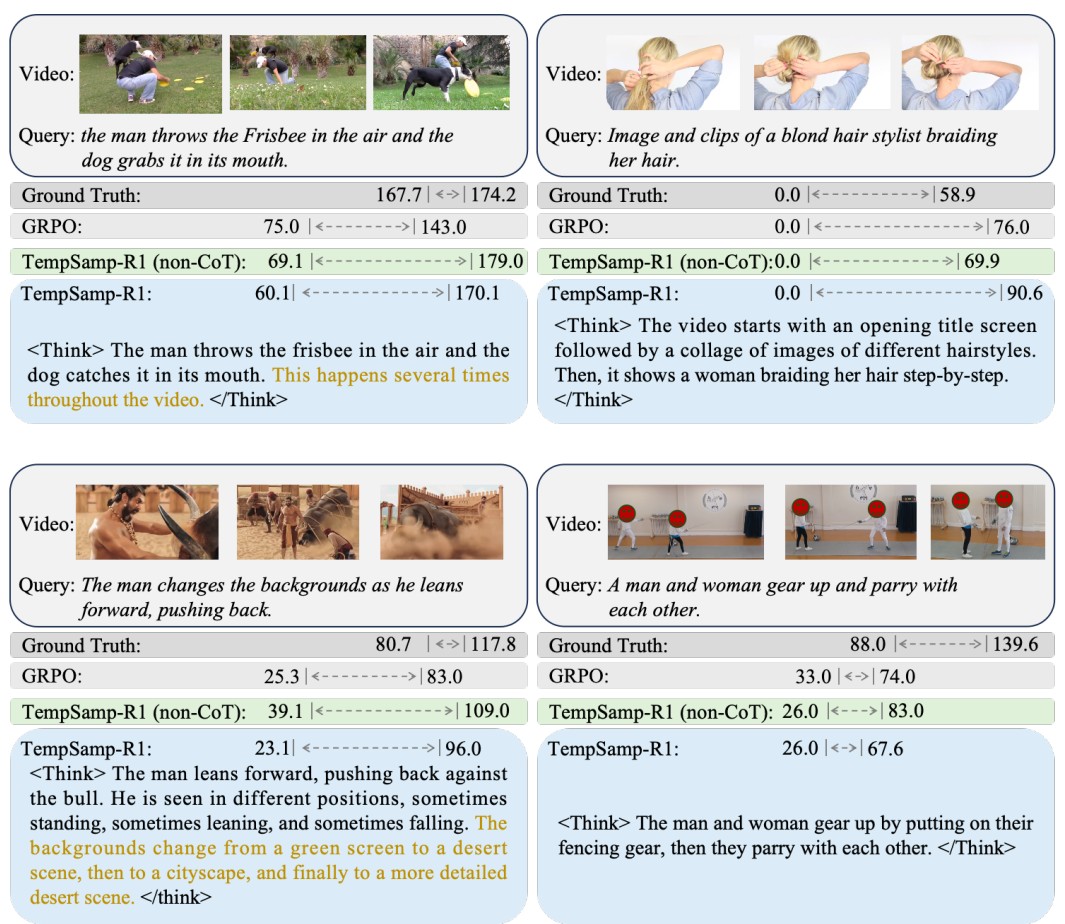

Figure 8: Qualitative analysis of failure cases. Most prediction errors arise from ambiguous visual cues or imperfect ground-truth annotations, such as repetitive events and ill-defined temporal boundaries.

Temporal video grounding (Non-CoT):

"To accurately pinpoint the event "query" in the video, determine the precise time period of the event. Provide the start and end times (in seconds, precise to two decimal places) in the format "start time to end time" within the <answer> </answer> tags. For example: "12.54 to 17.83"."

Temporal video grounding (CoT):

"To accurately pinpoint the event "query" in the video, determine the precise time period of the event. Output your thought process within the <think> </think> tags, including analysis with either specific timestamps (xx.xx) or time ranges (xx.xx to xx.xx) in <timestep> </timestep> tags. Then, provide the start and end times (in seconds, precise to two decimal places) in the format "start time to end time" within the <answer> </answer> tags. For example: "12.54 to 17.83"."

Video highlight detection (Non-CoT):

"Please find the highlight contents in the video described by a sentence query, determining the highlight timestamps and its saliency score on a scale from 1 to 5. Provide the highlight timestamps and its saliency score within the <answer> </answer> tags. The output format should be like: 'The highlight timestamps are in the 82, 84, 86, 88, 90, 92, 94, 96, 98, 100 seconds. Their saliency scores are 1.3, 1.7, 1.7, 1.7, 1.7, 1.3, 1.7, 2.3, 2.3, 2.3'. Now I will give you the sentence query: 'query'. Please return the query-based highlight timestamps and salient scores."

Video highlight detection (CoT):

"Please find the highlight contents in the video described by a sentence query, determining the highlight timestamps and its saliency score on a scale from 1 to 5. Output your thought process within the <think> </think> tags, including analysis with either highlight timestamps. Then, provide the highlight timestamps and its saliency score within the <answer> </answer> tags. The output format should be like: 'The highlight timestamps are in the 82, 84, 86, 88, 90, 92, 94, 96, 98, 100 seconds. Their saliency scores are 1.3, 1.7, 1.7, 1.7, 1.7, 1.3, 1.7, 2.3, 2.3, 2.3'. Now I will give you the sentence query: 'query'. Please return the query-based highlight timestamps and salient scores."

Figure 9: Prompt templates for temporal grounding and video highlight detection tasks.

