# OpenReview forum: "TempSamp-R1: Effective Temporal Sampling with Reinforcement Fine-Tuning for Video LLMs"
_NeurIPS.cc/2025/Conference — NeurIPS 2025 poster_

### Official Review · Reviewer_sPEX · 2025-06-09

**Clarity:** 3
**Significance:** 2
**Originality:** 2
**Rating:** 5
**Confidence:** 2

**Summary:**

This work proposes a method based on Group Relative Policy Optimization (GRPO) to fine-tune multimodal large language models (MLLMs ) on video grounding tasks. On top of GRPO, which has already been proven beneficial for video understanding and grounding tasks, this method introduces off-policy guidance, which further stabilizes the optimization process. The proposed method is evaluated on three dataset and tested against state-of-the-art models, both GRPO-based and using standard fine-tuning.

**Questions:**

**Q1:** **Table 1 is missing several important numbers and metrics**. While I understand that some data may not be available in the original papers, the table loses valuable information as it currently stands. For most of the models, certain metrics have been reported; therefore, when available, a trained model can be evaluated to provide the missing values. For example, if TempSamp-R1 can not be evaluated on other RL-based approaches on QVHighlights, then the comparison on this dataset is weak.
Additionally, I do not understand whether the *Qwen2.5-VL-7B* numbers refer to a zero-shot evaluation of the model or if they involve some level of fine-tuning. If they do not (as I assume from the -), can you explain why these numbers differ from those presented in the original paper [3]?

**Q2:** Why are the **SFT and GRPO baselines** missing from Table 1? I see results in the ablation studies and the OOD experiments, but I think it is important to compare the differences across all datasets considered.

**Q3:** It is not clear to me the real **contribution** of this paper. It seems that most of the mechanisms used were already used in previous work (GRPO for VTG task, iou reward, format reward, \<think\>). Is the main contribution only replacing random on-policy sampling with off-policy solutions to alleviate reward sparsity? The value of this, if I am not mistaken, is shown only in Table 4 and for one dataset (see Q2). The authors should elaborate on this to make their contributions stand out.

**Q4:** Does the **first stage of training** use in-domain data or a different dataset? I think this information is missing in the current version of the paper.

**Q5:** This is a **comment** rather than a question. I noticed that the **TimeZero numbers** you reported do not align with the current version of the paper you cited, which was updated on May 26th. I understand that this is concurrent work; if you choose to cite it and include it in your comparison, you should update the numbers accordingly.

**Ethical Concerns:**

["NO or VERY MINOR ethics concerns only"]

**Final Justification:**

I had multiple questions, and my initial rating was below acceptance. The authors addressed my concerns and detailed the proposed modifications they will implement. Consequently, I think this paper meets the required standard, and I have decided to raise my score.

**Limitations:**

yes

**Paper Formatting Concerns:**

The captions for the tables should be placed above the tables rather than below them.

**Quality:**

3

**Strengths And Weaknesses:**

**Strengths:** The paper proposes a method using  Group Relative Policy Optimization (GRPO) to fine-tune multimodal large language models (MLLMs ) on video grounding tasks, i.e., video temporal video grounding and highlight detection. This method replaces random on-policy sampling with off-policy solutions, reducing reward sparsity. The proposed approach is tested on 3 datasets. The idea follows the recent trend of using GRPO for fine-tuning MLLMs for tasks requiring temporal understanding.

**Weaknesses:** While the approach obtains higher results on the baselines and previous GRPO-based approaches, it is tested on one model only (Qwen2.5-VL-7B) and shares part of the methodology with previous works (e.g., GRPO for VTG task, iou reward, format reward, \<think\>). As I detail in the question section below, I believe some numbers are missing from the reported tables, which weakens the overall evaluation.

**Minor Weaknesses:** There are three **methodname** left in the paper by mistake (L274, L276, L279).

---

> ### Author Rebuttal · Authors · 2025-07-30
>
> ## W1&Q2: One model only (Qwen2.5-VL-7B); missing SFT and GRPO baselines in Table 1.
>
> Thank you for the valuable question. We have added comparisons on both Qwen2.5-VL-7B and Qwen2.5-VL-3B, including full baselines (SFT and GRPO). As shown in the updated table, TempSamp-R1 consistently outperforms GRPO and SFT under both model sizes, highlighting the robustness of our method beyond a single backbone.
>
> Notably, GRPO struggles to provide stable gains on the weaker Qwen2.5-VL-3B model, particularly on QVHighlights, where its performance is comparable to SFT. We attribute this to the high variance and sparse reward signals in GRPO, which are more difficult to optimize for smaller models. TempSamp-R1 effectively mitigates this limitation through off-policy supervision and stabilized reward shaping.
>
> We will include these experimental results and analyses in the final version.
>
> ### Table: Results on Charades-STA Dataset
>
> | Model              | Method      | mIoU | R1@0.3 | R1@0.5 | R1@0.7 |
> | ------------------ | ----------- | ---- | ------ | ------ | ------ |
> | Qwen2.5-VL-7B      | -           | 29.0 | 43.9   | 24.2   | 11.1   |
> | Qwen2.5-VL-7B      | SFT         | 56.8 | 79.6   | 66.7   | 43.5   |
> | Qwen2.5-VL-7B      | GRPO        | 59.0 | 81.2   | 70.5   | 47.1   |
> | Qwen2.5-VL-7B      | TempSamp-R1 | **61.7** | **83.3**   | **73.6**   | **52.2**   |
> |             |                 |             |        |        |        |
> | Qwen2.5-VL-3B      | -           | 33.8 | 51.8   | 32.1   | 15.0   |
> | Qwen2.5-VL-3B      | SFT         | 46.1 | 68.6   | 51.5   | 26.5   |
> | Qwen2.5-VL-3B      | GRPO        | 48.7 | 72.4   | 55.3   | 29.4   |
> | Qwen2.5-VL-3B      | TempSamp-R1 | **54.8** | **78.2**   | **64.8**   | **38.3**   |
>
> ### Table: Results on ActivityNet Captions Dataset
>
> | Model              | Method      | mIoU | R1@0.3 | R1@0.5 | R1@0.7 |
> | ------------------ | ----------- | ---- | ------ | ------ | ------ |
> | Qwen2.5-VL-7B      | -           | 21.1 | 28.3   | 15.8   | 7.5    |
> | Qwen2.5-VL-7B      | SFT         | 39.6 | 54.5   | 37.4   | 23.0   |
> | Qwen2.5-VL-7B      | GRPO        | 47.2 | 68.6   | 47.3   | 26.9   |
> | Qwen2.5-VL-7B      | TempSamp-R1 | **52.1** | **72.8**   | **55.4**   | **34.2**   |
> |             |                 |             |        |        |        |
> | Qwen2.5-VL-3B      | -           | 17.0 | 24.2   | 12.3   | 5.3    |
> | Qwen2.5-VL-3B      | SFT         | 37.8 | 54.0   | 35.3   | 19.3   |
> | Qwen2.5-VL-3B      | GRPO        | 42.2 | 62.3   | 44.5   | 24.2   |
> | Qwen2.5-VL-3B      | TempSamp-R1 | **45.3** | **64.0**   | **46.3**   | **27.1**   |
>
> ### Table: Results on QVHighlights Dataset
>
> | Model              | Method      | mAP  | HIT@1 |
> | ------------------ | ----------- | ---- | ------ |
> | Qwen2.5-VL-7B      | -           | 19.7 | 34.1   |
> | Qwen2.5-VL-7B      | SFT         | 26.5 | 34.1   |
> | Qwen2.5-VL-7B      | GRPO        | 18.0 | 35.0   |
> | Qwen2.5-VL-7B      | TempSamp-R1 | **30.0** | **57.6**   |
> |             |                 |             |        |        |        |
> | Qwen2.5-VL-3B      | -           | 13.8 | 22.8   |
> | Qwen2.5-VL-3B      | SFT         | 22.8 | 28.7   |
> | Qwen2.5-VL-3B      | GRPO        | 10.9 | 14.9   |
> | Qwen2.5-VL-3B      | TempSamp-R1 | **27.9** | **41.0**   |
>
>
>
> ## W2&Q3: Clarifying the main contributions.
>
> Thank you for the clarification request. Prior work, including GRPO for VTG, IoU reward, format reward, and think prompting, **serves as the foundation of our baseline**. Building upon these components, our method addresses a central limitation in video temporal grounding: unstable policy optimization under **sparse** and **temporally misaligned** reward supervision.
>
> To address these limitations, our contribution lies in designing a more stable and generalizable reinforcement fine-tuning framework through two novel components:
>
> (1) Off-policy guidance, which for the first time incorporates high-quality, instruction-aligned supervision from ground-truth annotations directly into policy updates. Unlike prior GRPO-based methods that treat ground-truth solely as a reference for reward evaluation of on-policy samples, our approach leverages these annotations as active supervision signals, providing temporally precise guidance that improves stability and convergence during training.
>
> (2) Non-linear reward transformation, which dynamically reshapes sparse and skewed reward distributions to enable stable policy optimization. In contrast to GRPO, which treats reward values uniformly and is prone to instability under outlier-biased distributions, our transformation preserves informative gradients from partially correct predictions while mitigating the influence of extreme values.
>
> (3) As shown in Table 4, our ablation and comparative studies demonstrate the complementary effects of both components. In addition, we extend the evaluation in W1 across multiple datasets (Charades-STA, ActivityNet Captions, QVHighlights) and model scales (Qwen2.5-VL-3B and 7B), confirming the effectiveness and generalizability of our method.
>
> We will clarify this distinction more explicitly in the final version.
>
>
> ## W3&Q1: Missing values and incomplete comparisons; Qwen2.5-VL-7B setup in Table 1.
>
> We appreciate the reviewer’s observation. As shown in the table in W1, we have added several missing metrics, including GRPO, SFT and baseline results, to support a more complete and fair comparison. Additionally, QVHighlights is a key benchmark for temporal grounding, yet was omitted in prior RL-based methods. To address this gap, we have included GRPO baselines on this dataset and extended the evaluation to multiple model sizes, demonstrating the effectiveness and generalizability of TempSamp-R1.
>
> The reported results for Qwen2.5-VL are from zero-shot evaluation without any fine-tuning. For fair comparison with TimeZero and VideoChat-R1, we standardized the total number of input pixels per video to approximately 2.8 million, which may lead to slight discrepancies from the original paper. We will clarify this setup in the final version.
>
> ## Q4: Clarification on training data for the first stage.
>
> Thank you for the question. The first-stage fine-tuning uses the same in-domain datasets as the second stage, matching the corresponding evaluation benchmarks (e.g., Charades-STA and QVHighlights). We will clarify this detail in the final version.
>
> ## Q5: TimeZero discrepancy.
>
> We thank the reviewer for noting this discrepancy. We had already noted the May 26th update and will revise the TimeZero results accordingly, clarifying the cited version in the final revision.
>
>
> ## Formatting concerns.
>
> Thank you for pointing this out. We will fix the placeholders and move all table captions above the tables.
>
>
> ### **Please feel free to leave additional comments if you have further questions or suggestions.**

---

> > ### Comment · Reviewer_sPEX · 2025-08-01
> >
> > Thank you for your response.
> >
> > I have reviewed the other reviewers’ comments as well as the authors’ replies. While some of my questions have been addressed (**Q2**, **Q4**, **Q5**), I believe others still warrant further clarification.
> >
> >
> > Regarding the zero-shot evaluation of Qwen2.5-VL-7B (**Q1**), I believe there may be an error in your reported results. I raised this point in the first review stage because I wanted to be sure about the correctness of your experimental setup.
> >
> > The mIoU you report on Charades-STA for Qwen2.5-VL-7B is 29.0, which is significantly lower than the 43.6 reported in the original paper. **I do not think such a large discrepancy can be attributed to a slight difference in the number of frames used, as suggested in your response.**
> >
> > Additionally, your mention of limiting input pixels per video to approximately 2.8 million is unclear. Could you please specify what this means in terms of resolution and number of frames in your evaluation, especially in comparison to the original model's setup? Without this information, it's difficult to assess the validity and fairness of your experimental design.
> >
> >
> > > We will clarify this distinction more explicitly in the final version.
> > >
> >
> > Please **specify exactly what content you intend to revise or add**, and where in the paper this clarification will appear. A vague statement is not sufficient. You need to clearly indicate what details will be added or changed, as these modifications will be considered part of the final version if the paper is accepted. Without this specificity, it is difficult to assess the seriousness and completeness of the proposed revisions.

---

> ### Author Response · Authors · 2025-08-03
> **Response to Reviewer sPEX: Clarifying Evaluation Settings**
>
> Thank you for your response. We appreciate the reviewer’s careful evaluation and constructive feedback.
>
> ## A1: Regarding Qwen2.5-VL-7B zero-shot evaluation
>
> Thank you for raising this important point. After rechecking our setup, we confirm that the discrepancy with the original paper arises from two distinct evaluation settings.  The table below reports the scores under both settings, illustrating that the difference is primarily due to the total_pixels and prompt ordering.
>
> - TimeZero / VideoChat-R1–aligned setting: For fair and computationally feasible comparison with TimeZero and VideoChat-R1, we standardized the total_pixels input to ≈2.8M per video and adopted the “Query+Video” prompt order. In practice, this corresponds to ≈86 frames at 168×336 resolution with FPS=2 (total visual input ≈2 × 2.8M pixels due to the initial 2×3×3 convolution used for visual embedding in Qwen2.5-VL). This lower input resolution and prompt design reduce the zero-shot score.
>
> - lmms-eval–aligned setting (original model reference): The original Qwen2.5-VL-7B evaluation in lmms-eval uses ≈19M total_pixels per video at FPS=2 (≈86 frames at 476×868 resolution), and the “Video+Query” prompt order. For example, a typical prompt is structured as: <|vision_start|><|video_pad|><|vision_end|>To accurately pinpoint the event "a person is standing in their entryway undressing." in the video, determine the precise time period of the event. Provide the start and end times (in seconds, precise to two decimal places) in the format "start time to end time" within the \<answer> \</answer> tags. For example: "12.54 to 17.83".
>
>
> ### Table: Results on Charades-STA Dataset
>
> |               | total_pixels | prompt      | mIoU | R1@0.3 | R1@0.5 | R1@0.7 |
> | ------------- | ------------ | ----------- | ---- | ------ | ------ | ------ |
> | Qwen2.5-VL-7B | 19267584     | Video+Query | 49.7 | 73.4   | 54.4   | 30.3   |
> | Qwen2.5-VL-7B | 2809856      | Video+Query | 46.6 | 71.2   | 49.5   | 25.5   |
> | Qwen2.5-VL-7B | 19267584     | Query+Video | 32.2 | 49.3   | 30.0   | 13.7   |
> | Qwen2.5-VL-7B | 2809856      | Query+Video | 29.0 | 43.9   | 24.2   | 11.1   |
>
> We also re-evaluated ActivityNet Captions and QVHighlights under the lmms-eval–aligned setting. We observed that the performance on QVHighlights is lower than under the TimeZero–aligned setting.
>
> ### Table: Results on ActivityNet Captions Dataset
>
>
> |               | mIoU                 | R1@0.3        | R1@0.5        | R1@0.7        |
> | ------------- | ------------ | ----------- | ---- | ----- |
> | Qwen2.5-VL-7B | 33.1                 | 45.2          | 29.7          | 18.1          |
>
> ### Table: Results on QVHighlights Dataset
>
> |               |               | mAP  | HIT@1 |
> | ------------- | ------------- | ---- | ----- |
> | Qwen2.5-VL-7B | TimeZero-aligned | 19.7 | 34.1  |
> | Qwen2.5-VL-7B | lmms-eval-aligned | 16.6 | 31.2  |
>
> Following the lmms-eval–aligned setting, we re-trained and evaluated both GRPO and our TempSamp-R1 method using a 2.8M total_pixel input per video and the “Video+Query” prompt format. As shown in the table, both prompt configurations (Query+Video vs. Video+Query) yield comparable performance, and TempSamp-R1 consistently outperforms GRPO under either setting. These results further confirm the effectiveness and robustness of our method across different prompt designs.
>
> ### Table: Results on Charades-STA Dataset
>
> |            | prompt      | mIoU | R1@0.3 | R1@0.5 | R1@0.7 |
> | ---------- | ----------- | ---- | ------ | ------ | ------ |
> | GRPO       | Query+Video | 59.0 | 81.2   | 70.5   | 47.1   |
> | TempSamp-R1 | Query+Video | 61.7 | 83.3   | 73.6   | 52.2   |
> |            |             |      |        |        |        |
> | GRPO       | Video+Query | 59.1 | 81.6   | 70.2   | 46.3   |
> | TempSamp-R1 | Video+Query | 60.6 | 82.6   | 72.3   | 50.5   |
>
> In summary, we will make the following updates in the final version for clarity:
> - Update Table 1 to report the Qwen2.5-VL results under their corresponding evaluation settings, with explicit annotation of the evaluation protocol and prompt order for each dataset. Charades-STA and ActivityNet Captions are evaluated under the lmms-eval–aligned setting using the Video+Query prompt. QVHighlights is evaluated under the TimeZero–aligned setting using the Query+Video prompt.
> - Explicitly indicate that the results for TimeZero, VideoChat-R1, and TempSamp-R1 follow the TimeZero–aligned setting to ensure fair comparison.
> - Provide additional results for GRPO and TempSamp-R1 trained and evaluated under the lmms-eval–aligned setting, so that the performance under the original evaluation protocol is also fully reported.

---

> ### Author Response · Authors · 2025-08-03
> **Response to Reviewer sPEX: Planned revisions**
>
> ## A2: Planned revisions for the final version.
> We thank the reviewer for the constructive feedback and for highlighting the need to make our contributions and distinctions from prior GRPO-based methods more explicit. In the final version, we will incorporate the following revisions to ensure clarity and completeness:
>
> 1. Introduction: We will insert a concise paragraph explicitly articulating our core contribution: “Building upon GRPO, the core contribution of TempSamp-R1 lies in integrating off-policy ground-truth annotations as active supervision within policy optimization, and employing a non-linear reward transformation to stabilize learning under sparse and temporally misaligned rewards.”
>
> 2. Method: In Sec. 3.2 (Mixed-policy sampling), we will add a clarifying sentence before the current description of advantage computation: “In contrast to prior GRPO-based methods, which utilize ground-truth only for reward evaluation of on-policy samples, TempSamp-R1 incorporates these annotations directly into advantage estimation to actively guide policy updates.”
>
> 3. Method: In Sec. 3.2 (Non-linear Reward Transformation), we will add a clarifying sentence at L175: “The proposed non-linear transformation preserves informative gradients from partially correct predictions and mitigates the dominance of outlier rewards, thereby enhancing training stability under sparse and skewed reward distributions.”
>
> 4. Experiments: We will supplement Table 1 with complete baselines, including SFT, GRPO, and our TempSamp-R1 across different model scales (Qwen2.5-VL-3B and 7B). We will also add the following analysis: “The results demonstrate that TempSamp-R1 consistently outperforms SFT and GRPO across model scales, highlighting the robustness and generalizability of our framework.”
>
>
> ### **We welcome any further comments or suggestions that could help improve the clarity and completeness of the final version.**

---

> > ### Comment · Reviewer_sPEX · 2025-08-04
> >
> > Thank you.
> > My concerns have been adequately addressed. I believe the additional analysis discussed during this period will strengthen the paper's submission. I will discuss the score with the other reviewers, but I am inclined to raise my rating.

---

> > > ### Author Response · Authors · 2025-08-04
> > > **Response to Reviewers:  Strengthening the paper**
> > >
> > > Thank you for the positive feedback. We sincerely appreciate your acknowledgment that our clarifications and additional analyses have addressed the concerns. We are pleased that these revisions will further strengthen the paper.
> > >
> > > If there are any additional points or clarifications that could assist in raising the rating, we would be pleased to provide them promptly.

---

### Official Review · Reviewer_J4MX · 2025-06-19

**Clarity:** 4
**Significance:** 3
**Originality:** 3
**Rating:** 5
**Confidence:** 4

**Summary:**

The paper introduces TempSamp-R1, a reinforcement fine-tuning framework that enhances the temporal grounding capabilities of multimodal large language models (MLLMs) for video understanding. It addresses the inefficiencies of prior methods like GRPO by incorporating off-policy supervision (i.e., ground-truth annotations) and non-linear soft advantage estimation to improve stability and exploration during training. The authors further integrate Chain-of-Thought (CoT) prompting into a unified model to support both CoT and non-CoT inference. TempSamp-R1 achieves state-of-the-art results on Charades-STA, ActivityNet Captions, and QVHighlights, and demonstrates strong performance in few-shot and cross-domain settings.

**Questions:**

See Weakness.

**Ethical Concerns:**

["NO or VERY MINOR ethics concerns only"]

**Final Justification:**

Thanks authors for their responses. My concerns have been addressed and I raised my score.

**Limitations:**

See Weakness.

**Paper Formatting Concerns:**

NA.

**Quality:**

4

**Strengths And Weaknesses:**

Strengths:

1.	The paper critically analyzes why GRPO struggles with sparse rewards and unstable updates in large temporal search spaces. Based on this, a novel method is proposed to incorporate groud-truth annotations during RL training. This method is strengthen by a non-linear reward shaping to stabilizes the training.

2.	The proposed method achieves generally better performance on multiple temporal grounding benchmarks. Futhur, the method demonstrates advantage over GRPO on cross-domain generalization and few-shot experiments. Extensive analysis and ablation studies also facilitate the intake of proposed method.

3.	The writing is generally great by demonstrating the intuition and understanding behind the proposed method.

Weaknesses:

1.	Question about off-policy output: As shown in Figure 2, during training, the model should be forced to output the ground-truth prediction so that the model weights can be updated with respective to this prediction. How was the model forced to do that?

2.	Question about Eq. 4: Can authors justify the design of this function? Specifically, why it has to be the format as Eq. 4 rather than other alternatives? Is it simply the best formulation from a few empirical observations?

3.	TempSamp-R1 (CoT): In L180-183, the model is trained in two stages where both stages are RL training. Since there is no CoT annotations used, how can we expect the model to output reliable CoT in the second stage?

4.	TempSamp-R1 Mixed CoT: In L234-235, how were the “better output” selected from two models? Did authors simply choose one of the two outputs that achieve higher scores of grounding metrics?

5.	Table 3: The proposed method requires much less time than GRPO. How does the proposed method save training time?

6.	Figure 3: How was the skewness score computed?

7.	The necessity of CoT: Even though CoT achieve better results (Table 6: +0.6 mIoU between row 5 and row 8), it incurs much more computation during inference. Is it necessary for temporal grounding, given that the improvement is not significant?

---

> ### Author Rebuttal · Authors · 2025-07-30
>
> ## Q1: How is ground-truth used for off-policy supervision?
> Thank you for the clarification request. The model is not forced to output the ground-truth prediction. Instead, ground-truth annotations are incorporated into the advantage estimation as external supervision signals to stabilize learning. The policy itself is still optimized based on gradients computed from solutions and their corresponding advantage values. We will clarify this distinction more explicitly in the final version.
>
> ## Q2: Justification for the functional form of Eq. 4.
>
> Thank you for the insightful question. In video temporal grounding, rewards are often sparse and temporally misaligned, making it difficult for partially correct predictions to receive useful feedback, especially when incorporating off-policy guidance that may diverge from the policy distribution. To address this, Eq. 4 is motivated by self-adaptive reward shaping techniques that have been widely applied to stabilize learning. We adopt an asymmetric transformation that compresses the advantages of near-optimal outputs while amplifying differences among suboptimal ones. This encourages the model to focus on high-IoU outputs while still benefiting from partially correct predictions, thereby improving the quality of gradient feedback. We will clarify the design rationale and task-specific motivations more explicitly in the final version.
>
> ## Q3: How can the model learn reliable CoT outputs?
>
> Thank you for the thoughtful question. Rather than explicitly enforcing CoT generation, our framework treats CoT as an adaptive strategy that the model may utilize when beneficial for task performance. First, the base Qwen2.5-VL model already demonstrates CoT-style reasoning, and this ability is preserved after the first training stage. During the second stage, the model learns through reward-guided exploration that producing intermediate reasoning steps can improve grounding rewards. Representative examples of such outputs can be found in Figures 5, 6, and 7. We will clarify this design intent more explicitly in the final version.
>
> ## Q4: How is the “better output” selected in Mixed CoT mode?
>
> In Table 1, TempSamp-R1 Mixed CoT simply selects the output (CoT or non-CoT) with the higher grounding score to illustrate their complementarity. We also experimented with simple heuristics such as using video length for selection, but found no consistent advantage over single-mode inference. We will clarify the selection strategy more explicitly in the final version.
>
> ## Q5: Why is TempSamp-R1 more time-efficient than GRPO?
>
> TempSamp-R1 achieves higher time efficiency primarily through two factors. First, our off-policy guidance allows the direct incorporation of ground-truth solutions into advantage estimation, which reduces the number of on-policy samples required per query from $G$ (in GRPO) to $G{-}1$, thereby lowering the computational cost of sampling. Second, we observe that our method produces shorter average completions during training (78 vs. 118 tokens), leading to reduced sequence generation overhead. We will clarify both aspects of this efficiency gain in the final version.
>
> ## Q6: How is the skewness score computed?
>
> We compute skewness using the standard third standardized moment:
>
> $S = \frac{1}{G} \sum_{i=1}^G \left( \frac{r_i - \mu}{\sigma} \right)^3$,
>
> where $r_i$ is the reward of the $i$-th sample, $\mu$ and $\sigma$ are the mean and standard deviation of the reward distribution, and $G$ is the total number of solutions. This computation is implemented using scipy.stats.skew, and it quantifies the asymmetry and sharpness of the reward distribution during training. We will clarify this in the final version.
>
>
> ## Q7: The necessity of CoT
>
> Thank you for the valuable question.  In our experiments conducted on a single NVIDIA A100 GPU, the inference time for CoT outputs remains comparable to that of non-CoT outputs (2.12s vs. 2.10s per sample), due to the relatively short length of generated reasoning steps. Furthermore, our framework is designed to accommodate both CoT and non-CoT reasoning within a unified model. As shown in Table 1, we find that CoT can improve grounding quality with minimal computational overhead in some settings (e.g., Charades-STA), making it a viable option when quality gain justifies the cost. However, in tasks where CoT offers limited performance benefits (e.g., QVHighlights), it can be omitted to reduce latency. We will clarify this point in the final version.
>
> ### **Please feel free to leave additional comments if you have further questions or suggestions.**

---

### Official Review · Reviewer_xy81 · 2025-07-02

**Clarity:** 4
**Significance:** 3
**Originality:** 2
**Rating:** 4
**Confidence:** 3

**Summary:**

This paper introduces a reinforcement fine-tuning framework to improve multimodal large language models (MLLMs) for video temporal grounding. Due to the vast search space for time intervals in this task, methods like GRPO can be inefficient. To address this, TempSamp-R1 incorporates ground-truth annotations as off-policy supervision, adding one off-policy solution alongside on-policy samples per query. Additionally, to prevent instability from directly using off-policy rewards, the paper proposes three strategies—reward downscaling, advantage anchoring, and non-linear reward shaping—to stabilize training. Experimental results in video temporal grounding benchmarks validate improvements over GRPO-based baselines.

**Questions:**

Overall, it would be valuable for this paper to provide more experimental clarification on why the proposed method works well—whether it benefits from CoT, normalization, or other factors.

Q1. [See W1]

Q2. [See W2]

Q3. **Limited Exploration vs. Stable Convergence Trade-off:** The reduced variance in Top-1 IoU rewards for TempSamp-R1 in Figure 4 is presented as evidence of stability. However, could this tighter distribution also indicate reduced exploration of the solution space compared to the GRPO baseline? While the median reward is higher, is there a risk that TempSamp-R1 may converge to a local optimum, potentially limiting its ability to discover diverse or novel high-quality solutions?

**Ethical Concerns:**

["NO or VERY MINOR ethics concerns only"]

**Final Justification:**

My previous concerns on the ablation study and the parameter selection have been resolved during the rebuttal and discussion phase. Therefore I maintain my initial score, 4: borderline accept, and lean towards acceptance.

**Limitations:**

yes

**Quality:**

3

**Strengths And Weaknesses:**

## Strengths

S1. **Clear Presentation:** The paper is well-written and easy to follow, with clear explanations that make the context easy to understand.

S2. **Clear Motivation:** The motivation is well-argued. As shown in Figure 4, while GRPO’s ability to enrich the search space with on-policy solutions is beneficial, it can also hinder the model's training stability. This is reasonable, given that pretrained MLLMs may not have sufficient knowledge of temporal grounding tasks, making it hard to generate helpful signals, especially at the beginning of training.

S3. **Performance**: Experimental results (Table 1) shows that the complete pipeline clearly outperforms other SFT methods or RL methods using the same model in video temporal grounding tasks.



## Weaknesses

W1. **Need Clarification on Contribution (Reward design vs. CoT):** I believe the strength of GRPO in reasoning tasks lies in its ability to prompt MLLMs for additional thinking steps, even without explicitly teacher-forcing intermediate steps based on ground-truth. In this regard, I am curious about the difference between GRPO (CoT) and TempSamp-R1 (CoT). The ablation study only compares GRPO (non-CoT), TempSamp-R1 (non-CoT), and TempSamp-R1 (CoT). I wonder whether the observed improvements come primarily from the stabilized reward design, from the use of CoT itself, or the synergy of both.

W2. **Heuristics on Parameters:** The specific non-linear function (Equation 4) and its parameters ($\tau$ = 0.8, $\alpha_1$ = 0.01, $\alpha_2$ = 1) appear to be empirically chosen heuristics. While it is common practice to carefully select hyper-parameters for stable RL fine-tuning, I am curious about how sensitive the proposed design is to these parameters.

---

> ### Author Rebuttal · Authors · 2025-07-30
>
> ## Q1&W1: Clarifying the contribution of CoT vs. reward design.
>
> Thank you for the insightful question. We conducted additional comparative experiments between GRPO and TempSamp-R1 under both CoT and non-CoT prompting strategies. The results confirm that TempSamp-R1 consistently outperforms GRPO across all settings, with or without CoT. Notably, the performance gains observed in the CoT setting (+1.5 R1@0.5 and +4.8 R1@0.7) are comparable to or even larger than those in the non-CoT setting (+1.4 R1@0.5 and +3.7 R1@0.7). This suggests that CoT prompting benefits more from stabilized training, allowing the model to more effectively exploit intermediate reasoning steps. Additionally, these results indicate that CoT prompting and the stabilized reward optimization, achieved through the combination of off-policy guidance and non-linear reward transformation, contribute complementary benefits. We will include these comparative results and further discussion in the final revision.
>
> | Method      | Training Prompt | Test Prompt | R1@0.3 | R1@0.5 | R1@0.7 |
> |-------------|-----------------|-------------|--------|--------|--------|
> | GRPO        | Non-CoT       | Non-CoT   | 81.2   | 70.5   | 47.1   |
> | TempSamp-R1 | Non-CoT       | Non-CoT   | **82.6**   | **71.9**   | **50.8**   |
> |             |                 |             |        |        |        |
> | GRPO        | CoT             | CoT         | 83.0   | 72.6   | 48.1   |
> | TempSamp-R1 | CoT             | CoT         | **83.6**   | **74.1**   | **52.9**   |
>
>
> ## Q2&W2: Heuristic parameter selection and sensitivity.
>
> Thank you for the thoughtful question.
> In video temporal grounding, rewards are often sparse and temporally misaligned, making it difficult for partially correct predictions to receive useful feedback, especially when incorporating off-policy guidance that may diverge from the policy distribution. To address this, Eq. 4 is motivated by self-adaptive reward shaping techniques that have been widely applied to stabilize learning. We adopt an asymmetric formulation to better distinguish partially correct outputs from suboptimal ones.
>
> We have added corresponding ablation experiments. Results show that the model is relatively insensitive to $\alpha_1$ and $\alpha_2$ within reasonable ranges (e.g., $\alpha_1 \in [0.01, 0.02]$, $\alpha_2 \in [0.5, 1]$). Regarding $\tau$, values near 0.8 generally led to improved training stability and performance in our experiments. Higher values (e.g., 0.9) tended to exacerbate reward imbalance by concentrating reward mass on very few samples, which limits the model's ability to benefit from informative but suboptimal solutions.
>
> We will include these ablation results and analyses in the final revision.
>
> | Method                  |  $\tau$     | R1@0.3 | R1@0.5 | R1@0.7 |
> |-------------------------|-------|--------|--------|--------|
> | non-linear reward shaping | 0.6   | 81.3   | 69.6   | 47.6   |
> | non-linear reward shaping | 0.7   | 81.6   | 70.5   | 48.4   |
> | non-linear reward shaping | **0.8**   | **82.9**   | **72.1**   | **49.6**   |
> | non-linear reward shaping | 0.9   | 81.1   | 69.4   | 47.5   |
>
> | Method                  | $\alpha_1$ | R1@0.3 | R1@0.5 | R1@0.7 |
> |-------------------------|----------|--------|--------|--------|
> | non-linear reward shaping | 0.05     | 81.6   | 70.5   | 48.4   |
> | non-linear reward shaping | **0.01**     | **82.9**   | **72.1**   | **49.6**   |
> | non-linear reward shaping | 0.02     | 81.9   | 71.0   | 49.1   |
>
> | Method                  | $\alpha_2$ | R1@0.3 | R1@0.5 | R1@0.7 |
> |-------------------------|----------|--------|--------|--------|
> | non-linear reward shaping | 0.5      | 82.2   | 71.6   | 49.0   |
> | non-linear reward shaping | **1**        | **82.9**   | **72.1**   | **49.6**   |
> | non-linear reward shaping | 2        | 81.1   | 69.5   | 46.9   |
>
>
> ## Q3: Trade-off between exploration and stable convergence.
>
> Thank you for the valuable question. As figures cannot be included in the rebuttal, we instead report policy entropy values at different training steps. Compared to GRPO, TempSamp-R1 demonstrates higher policy entropy on ActivityNet, suggesting stronger exploratory behavior during training. This effect arises in part from the use of ground-truth annotations as off-policy guidance, since these solutions typically fall outside the model's high-probability prediction region. Integrating such informative but underexplored outputs into the learning process enables the model to expand its exploration space without compromising convergence stability. We will include the relevant results and analysis in the final revision.
>
>
> ### Table: Entropy per Step for Policy Models (GRPO & TempSamp-R1)
>
> | Method/step     | 500   | 1000  | 2000  | 3000  | 4000  |
> |------------------|-------|-------|-------|-------|-------|
> | GRPO   | 0.775 | 0.730 | 0.685 | 0.675 | 0.647 |
> | TempSamp-R1 | 0.805 | 0.788 | 0.786 | 0.783 | 0.779 |
>
> ### **Please feel free to leave additional comments if you have further questions or suggestions.**

---

> > ### Comment · Reviewer_xy81 · 2025-08-04
> >
> > I sincerely thank the authors for their thorough responses and additional experiments. My concerns regarding individual contribution, parameter sensitivity, and exploration trade-offs have been adequately addressed. I believe the additional experimental results will enhance the reliability and validation of the experiments once added into the manuscript. After reviewing the other reviews and responses, I find more positives than negatives. Therefore, I will maintain my score and lean towards acceptance.

---

> > > ### Author Response · Authors · 2025-08-05
> > > **Appreciation for the Reviewer’s Recognition**
> > >
> > > We sincerely thank the reviewer for the thorough evaluation, positive feedback, and recognition of our method. We appreciate that our responses and additional experiments have addressed your concerns. We will incorporate your constructive comments into the final version to further enhance the paper’s clarity and presentation.

---

### Official Review · Reviewer_hB8a · 2025-07-05

**Clarity:** 3
**Significance:** 2
**Originality:** 3
**Rating:** 4
**Confidence:** 3

**Summary:**

This paper presents TempSamp-R1, a reinforcement learning framework aimed at improving the performance of multimodal large language models (MLLMs) in video temporal localization tasks. It addresses the inefficiencies of existing methods like GRPO by using ground-truth annotations for precise temporal guidance and introducing a nonlinear soft advantage mechanism to stabilize training. Experiments on benchmark datasets such as Charades-STA, ActivityNet Captions, and QVHighlights demonstrate that TempSamp-R1 outperforms GRPO-based approaches, particularly in few-shot scenarios.

**Questions:**

Overall, I am particularly interested in the **advantage anchoring** and **non-linear reward shaping** methods regarding their generalizability across other tasks (besides Temporal Grounding) and the robustness of their hyperparameters. It remains to be seen whether these methods are universally important when combined with **mixed-policy sampling** or if they are only effective specifically for **temporal grounding**. However, this does not constitute a negative opinion of the paper.

**Ethical Concerns:**

["NO or VERY MINOR ethics concerns only"]

**Limitations:**

yes.

**Paper Formatting Concerns:**

Not any.

**Quality:**

3

**Strengths And Weaknesses:**

Strengths
- Innovative Framework Design: TempSamp-R1 introduces a hybrid policy training strategy that combines off-policy guidance (e.g., ground-truth annotations) with on-policy generation to offer more accurate temporal supervision. This helps resolve the common issues of sparsity and misalignment in on-policy sampling. The effectiveness has been thoroughly demonstrated in the experimental parts.
- Stable Training Mechanism: By incorporating a nonlinear soft advantage estimation mechanism, TempSamp-R1 dynamically reshapes reward feedback, thereby reducing gradient variance and adjusting advantage bias. This facilitates robust exploration and convergence.
- Strong Experimental Results and Thorough Analysis: Experiments on multiple benchmark datasets show that TempSamp-R1 outperforms existing GRPO-based methods and exhibits strong generalization capabilities in few-shot scenarios.
- Clear Illustrations: The paper provides numerous figures, such as Fig. 3 and Fig.4 , which significantly improve both the readability the this paper and the interpretability of the proposed methods.

Weakness
- Both **advantage anchoring** and **non-linear reward shaping** are intuitive concepts. However, aside from a set of effectiveness experiments, the paper does not include ablation studies specifically targeting these two parameters.
- The article has significant practical value, but its overall innovativeness is somewhat limited.

---

> ### Author Rebuttal · Authors · 2025-07-30
>
> ## W1: Lack of ablations for advantage anchoring and non-linear reward shaping.
>
> Thank you for the valuable suggestion. We have added ablations for $\lambda_{\text{off}}$ (used in advantage anchoring), as well as $\tau$, $\alpha_1$, and $\alpha_2$ (hyperparameters within non-linear reward shaping).
> - We observe that a low $\lambda_{\text{off}}$ (e.g., 1.0) weakens off-policy supervision, while overly large values (e.g., 1.4) introduce distributional shifts between on- and off-policy solutions, leading to biased advantage estimation and degraded policy learning stability.
> - For non-linear reward shaping, the model shows robustness to $\alpha_1$ and $\alpha_2$ within reasonable ranges (e.g., $\alpha_1 \in [0.01, 0.02]$, $\alpha_2 \in [0.5, 1]$). Regarding $\tau$, values near 0.8 generally led to improved training stability and performance in our experiments. Higher values (e.g., 0.9) skew the reward distribution toward a few high-reward solutions, limiting the model’s ability to learn from suboptimal solutions.
>
> We will include these ablation results and analyses in the final revision.
>
> | Method           | $\lambda_{\text{off}}$ | R1@0.3 | R1@0.5 | R1@0.7 |
> |------------------|------|--------|--------|--------|
> | advantage anchoring | 1.0  | 81.9   | 69.7   | 48.5   |
> | advantage anchoring | **1.2**  | **81.8**   | **70.7**   | **49.1**   |
> | advantage anchoring | 1.4  | 81.8   | 68.8   | 46.5   |
>
> | Method                  |  $\tau$     | R1@0.3 | R1@0.5 | R1@0.7 |
> |-------------------------|-------|--------|--------|--------|
> | non-linear reward shaping | 0.6   | 81.3   | 69.6   | 47.6   |
> | non-linear reward shaping | 0.7   | 81.6   | 70.5   | 48.4   |
> | non-linear reward shaping | **0.8**   | **82.9**   | **72.1**   | **49.6**   |
> | non-linear reward shaping | 0.9   | 81.1   | 69.4   | 47.5   |
>
> | Method                  | $\alpha_1$ | R1@0.3 | R1@0.5 | R1@0.7 |
> |-------------------------|----------|--------|--------|--------|
> | non-linear reward shaping | 0.05     | 81.6   | 70.5   | 48.4   |
> | non-linear reward shaping | **0.01**     | **82.9**   | **72.1**   | **49.6**   |
> | non-linear reward shaping | 0.02     | 81.9   | 71.0   | 49.1   |
>
> | Method                  | $\alpha_2$ | R1@0.3 | R1@0.5 | R1@0.7 |
> |-------------------------|----------|--------|--------|--------|
> | non-linear reward shaping | 0.5      | 82.2   | 71.6   | 49.0   |
> | non-linear reward shaping | **1**        | **82.9**   | **72.1**   | **49.6**   |
> | non-linear reward shaping | 2        | 81.1   | 69.5   | 46.9   |
>
>
> ## W2: Significant practical value, with concerns regarding limited novelty.
> We appreciate the reviewer’s recognition of the practical value of our work. The effectiveness of our framework stems from addressing a **key limitation** in prior RL-based approaches (e.g.,TimeZero and VideoChat-R1) for video temporal grounding, specifically unstable policy learning under **sparse** and **temporally misaligned reward supervision**.
>
> To address these limitations, we propose two novel mechanisms:
>
> (1) Off-policy guidance. Our method introduces a novel use of ground-truth annotations by directly **integrating them into the policy update process**, rather than merely using them for **reward evaluation of on-policy samples** as in GRPO-based approaches. These instruction-aligned annotations serve as active supervision signals that enhance training stability by providing temporally accurate and semantically grounded feedback.
>
> (2) Non-linear reward transformation. To address the distributional bias introduced by incorporating off-policy rewards, we adopt a soft advantage estimation mechanism inspired by adaptive reward shaping. Rather than **treating all rewards uniformly**, our method applies an **asymmetric transformation** that compresses the advantage values of near-optimal solutions while amplifying distinctions among suboptimal ones. This strategy produces more informative gradients and mitigates the suppression of promising on-policy samples, thereby improving training stability and encouraging effective policy exploration.
>
> These components are seamlessly integrated within a CoT-compatible framework, resulting in consistent improvements over GRPO baselines across multiple dataset. We will make this contribution more explicit and emphasize its significance in the final version.
>
> ## Q1:  Generalizability and robustness across tasks.
>
> To assess generalizability beyond temporal grounding, we conducted additional experiments on the NExT-GQA benchmark following the training and evaluation setting introduced by VideoChat-R1. This task involves spatio-temporal grounding for video-based question answering, where the model must identify both the correct answer (acc) and its supporting temporal segment (mIoU).
>
> We observe that both advantage anchoring and non-linear reward shaping consistently outperform GRPO across multiple metrics. Notably, these results are achieved using the same hyperparameter configurations as in our temporal grounding experiments, without any task-specific tuning. This demonstrates that our proposed methods are robust and transferable to structurally different tasks that require temporal grounding with multimodal reasoning.
>
> |                          | NExTGQA |        |
> |--------------------------|---------|--------|
> |                          | mIoU    | acc    |
> | Qwen2.5-VL-7B        | 15.4    | 59.5   |
> | GRPO                     | 35.1    | 68.7   |
> | TempSamp-R1 with advantage anchoring | **35.5**    | **74.5**   |
> | TempSamp-R1 with non-linear reward shaping | **36.0**    | **76.7**   |
>
>
> ### **Please feel free to leave additional comments if you have further questions or suggestions.**

---

### Note · Authors · 2025-08-12

To the Area Chairs,

We sincerely thank all reviewers and the Area Chairs for their time, thoughtful evaluations, and constructive feedback throughout the review and discussion phases.

Our paper proposes TempSamp-R1, a reinforcement fine-tuning framework for improving video temporal grounding in multimodal large language models (MLLMs). Building upon GRPO, our key contributions are: (i) incorporating instruction-aligned ground-truth annotations directly into policy optimization as off-policy guidance, rather than solely using them for reward evaluation; and (ii) applying a non-linear reward transformation that reshapes sparse and skewed reward distributions to enable more stable and generalizable learning. These contributions address core limitations of GRPO in policy instability and sparse reward supervision and are validated across three datasets and two model scales (Qwen2.5-VL-3B and 7B).

**Importantly, all reviewers recognized the novelty, stability, and empirical effectiveness of our method, along with the clarity and quality of the paper’s writing.**

Reviewer hB8a highlighted our “innovative framework design,” “stable training mechanism,” and “strong experimental results.” and, though not active in discussion, maintained a positive overall assessment.

Reviewer xy81 emphasized the “clear motivation” and agreed that our method “clearly outperforms other SFT or RL methods.” and expressed a positive inclination toward acceptance.

Reviewer J4MX acknowledged our method as a “novel extension to GRPO” that integrates ground-truth supervision with improved reward shaping and expressed a positive inclination toward acceptance.

Reviewer sPEX acknowledged that the additional experiments addressed their concerns and expressed willingness to raise the score.


In our rebuttal, we systematically addressed all reviewer concerns, including ablation completeness (hB8a, xy81, J4MX), contributions beyond GRPO-based methods (hB8a, sPEX), model generalization (hB8a, xy81, sPEX), effect of CoT prompting (xy81, J4MX) and model details and baseline completeness (J4MX, xy81, sPEX). **All reviewers confirmed that their concerns were resolved and expressed increased positive support for acceptance following the discussion.** We have also proposed detailed revision plans to improve clarity, reproducibility, and experimental completeness in the final version.

Thank you for your time and consideration.

Best regards,

[Authors of Submission 3411]

---

### Decision · Program_Chairs · 2025-09-17

**Decision:**

Accept (poster)

**Comment:**

All reviewers recommend acceptance with ratings (4, 4, 5, 5). This work presents TempSamp-R1, a reinforcement learning framework designed to enhance the performance of multimodal large language models in video temporal localization tasks. Reviewers acknowledged the novelty of the proposed method, stable training mechanism, and strong empirical results. Initial concerns regarding specific ablations and the scope of the evaluation were well addressed by the authors in the rebuttal, which included new experiments that further strengthened the paper.  The AC panel recommends accept.